# Test-Time Accuracy-Cost Control in Neural Simulators via Recurrent-Depth

**Harris Abdul Majid***
The University of Edinburgh
h.abdulmajid@ed.ac.uk

**Pietro Sittoni**
Gran Sasso Science Institute

**Francesco Tudisco**
The University of Edinburgh
Miniml.AI

## ABSTRACT

Accuracy-cost trade-offs are a fundamental aspect of scientific computing. Classical numerical methods inherently offer such a trade-off: increasing resolution, order, or precision typically yields more accurate solutions at higher computational cost. We introduce **Recurrent-Depth Simulator** (**RecurrSim**) an architecture-agnostic framework that enables explicit test-time control over accuracy-cost trade-offs in neural simulators without requiring retraining or architectural redesign. By setting the number of recurrent iterations $K$, users can generate fast, less-accurate simulations for exploratory runs or real-time control loops, or increase $K$ for more-accurate simulations in critical applications or offline studies. We demonstrate RecurrSim's effectiveness across fluid dynamics benchmarks (Burgers, Korteweg-De Vries, Kuramoto-Sivashinsky), achieving physically faithful simulations over long horizons even in low-compute settings. On high-dimensional 3D compressible Navier-Stokes simulations with 262k points, a 0.8B parameter RecurrFNO outperforms 1.6B parameter baselines while using 13.5% less training memory. RecurrSim consistently delivers superior accuracy-cost trade-offs compared to alternative adaptive-compute models, including Deep Equilibrium and diffusion-based approaches. We further validate broad architectural compatibility: RecurrViT reduces error accumulation by 90% compared to standard Vision Transformers on Active Matter, while RecurrUPT matches UPT performance on ShapeNet-Car using 44% fewer parameters.

## 1 INTRODUCTION

Simulations are fundamental to science and engineering. They enable scientists to study and predict the behavior of complex systems, and engineers to quickly iterate and optimize designs, without the need for expensive or impractical experiments. Early scientific computing, limited by computational resources, produced crude simulations with limited practical value. Today, with the availability of enormous computers, simulations have led to breakthroughs across different domains, including numerical weather prediction, fluid and particle flow, and drug and material design. Still, even with today's computational resources, less accurate but fast simulations are essential for early-stage studies and prototyping.

In scientific computing, techniques for explicit control of the accuracy-cost trade-off are well-established. Heuristic search methods, such as genetic algorithms and simulated annealing, enable users to balance desired accuracy against available computational resources by controlling the size of the search space. For instance, genetic algorithms obtain better solutions with larger population sizes or by running more generations. Similarly, numerical methods, which have traditionally under-pinned practically all simulations, have an inherent accuracy-cost trade-off: using finer discretizations, higher-order methods, and lower tolerances yields more accurate solutions but requires more computational resources to execute. For high-dimensional or large-scale problems, this trade-off becomes extremely unfavorable, rendering many real-world problems computationally intractable.

Machine learning provides a promising avenue to overcome this trade-off. Unlike numerical methods, which rely on explicitly defined formulations or heuristics, machine learning methods are general-purpose learners that learn directly from the vast amounts of available measurement and observational data, and are capable of generating simulations for a wide range of problems, geometries, discretizations, and boundary conditions. Machine learning methods also benefit from hardware and software advancements specifically developed for the machine learning ecosystem, including GPU acceleration frameworks and automated parallelization tools. Additionally, in favorable settings, machine learning methods can achieve comparable accuracy with less computational resources, or deliver greater accuracy within the same computational budget. Several works have successfully demonstrated these advantages on challenging applications including atmosphere and weather forecasting and industrial automotive design (Price et al., 2025; Bleeker et al., 2025).

At train-time, there are a number of tunable knobs available for controlling the test-time accuracy-cost trade-off of a neural simulator. Generally, allocating more computational resources during training leads to more accurate predictions—whether that is by increasing the training dataset size through data acquisition, data augmentation, or synthetic data; by increasing the model size through stacking more layers or using wider layers; or by improving the optimization process through more advanced optimizers, higher numerical precision, or training for more steps. Each of these train-time adjustments directly affect the test-time accuracy and cost.

At test-time, once a neural simulator has been trained, there are fewer tunable knobs. Deep Equilibrium models can iterate for more steps by increasing the maximum iteration limit or lowering the convergence tolerance, with additional iterations theoretically yielding solutions closer to the true fixed point (Bai et al., 2019). Diffusion models can make use of additional denoising steps or more advanced samplers to generate higher-quality outputs at greater cost (Ho et al., 2020; Lu et al., 2022). Recent advances in large language model inference has developed adaptive computation methods that dynamically allocate computational resources based on input complexity, enabling fast inference on simple prompts while spending additional compute on more challenging prompts (Wei et al., 2022; DeepSeek-AI et al., 2025; Geiping et al., 2025).

Our contributions are as follows:

1. We introduce RecurrSim, an architecture-agnostic, plug-and-play framework that enables explicit test-time control over accuracy-cost trade-offs in neural simulators, allowing a single model to generate both fast simulations and high-accuracy simulations.

2. We validate RecurrSim across diverse datasets (Burgers, Korteweg-De Vries, Kuramoto Sivashinsky, Active Matter, and ShapeNet-Car) and architectures (FNO, ViT, UPT), demonstrating physically faithful simulations even in low-compute settings, with superior memory and parameter efficiency on high-dimensional problems.

3. We establish accuracy-cost curves as an evaluation framework for adaptive neural simulations, demonstrating that RecurrSim overcomes key limitations of existing methods: architecture dependence, early plateauing, lack of anytime prediction capability, poor generalization to OOD recurrent iterations, and parameter sensitivity.

## 2 BACKGROUND

**Partial Differential Equations.** We consider time-dependent partial differential equations of the form

$$\mathbf{u}_t + \mathcal{N}(t, \mathbf{x}, \mathbf{u}, \mathbf{u_x}, \mathbf{u_{xx}}, \dots) = 0,$$

where $t \in [0, T]$ represents the temporal dimension, $\mathbf{x} \in \mathcal{X}$ represents the (possibly multiple) spatial dimension(s), and $\mathbf{u}(t, \mathbf{x}) : [0, T] \times \mathcal{X} \to \mathbb{R}^n$ represents the state at $(t, \mathbf{x})$. Here, $\mathcal{N}$ is a nonlinear operator that governs the systems' dynamics, describing the interactions among the different variables and their derivatives. We consider initial conditions given by $\mathbf{u}(0, \mathbf{x}) = \mathbf{u}_0(\mathbf{x})$, and unless otherwise specified, assume periodic boundary conditions.

Discretizing the partial differential equations transforms the continuous equations into a discrete form, yielding a sequence of states at discrete time steps $\{\mathbf{U}_n\}_{n=0}^N$, where $N = T/\Delta t$ is the number of time steps $\Delta t$. This discretization induces an evolution operator $\mathcal{G}$, which maps the state at any given time step to the state at the subsequent time step $\mathcal{G} : \mathbf{U}_n \to \mathbf{U}_{n+1}$.

**Neural Simulators.** A neural (physics) simulator approximates the evolution operator $\mathcal{G}$ with a learned operator $\mathcal{G}_\theta$, often by minimizing the one-step loss $\mathcal{L} = ||U_{t+1} - \mathcal{G}_\theta(U_t)||_2^2$, using data from high-fidelity simulations or real-world measurements. Repeated application of $\mathcal{G}_\theta$ generates a trajectory. Because the one-step loss does not measure trajectory performance, accuracy is typically quantified by a *trajectory error*:

$$\frac{1}{N \cdot d} \sum_{n=1}^{N} \left\| U_n - \mathcal{G}_\theta^{(n)}(U_0) \right\|_2^2,$$

where $d$ denotes the number of spatial points and $\mathcal{G}_\theta^{(n)}$ denotes the $n$-fold application of the neural simulator. However, for chaotic systems where small errors grow exponentially, the trajectory error becomes unreliable as a measure of trajectory performance. Instead, we define

$$\tau_\alpha = \min\left\{ t = n\Delta t \,\big|\, \rho\left(\mathbf{U}_n, \mathcal{G}_\theta^{(n)}(\mathbf{U}_0)\right) < \alpha \right\},$$

to be the earliest time at which the Pearson correlation coefficient $\rho$ between the true and predicted state falls below a specified threshold $\alpha \in (0,1)$. Computing $\tau_\alpha$ for each test trajectory yields (i) the *average correlation horizon*, obtained by averaging all $\tau_\alpha$ values, and (ii) the *worst-case correlation horizon*, obtained by selecting the minimum $\tau_\alpha$. Together, the trajectory error and correlation horizons capture both long-term accuracy and stability.

**Related Work.** A wide range of architectures have been explored for neural simulators. For regular domains, convolutional-based architectures such as the Residual Network (ResNet (He et al., 2016)) and the U-shaped Encoder-Decoder (UNet (Ronneberger et al., 2015)) effectively capture local interactions, whereas spectral-based architectures, such as the Fourier Neural Operator (FNO (Li et al., 2020)) and its factorized variant (F-FNO (Tran et al., 2021)), leverage global frequency-domain features. For irregular domains, Brandstetter et al. (2022) propose a message-passing graph neural network, while Li et al. (2023) extend the FNO architecture with a geometry encoder and decoder, deforming an irregular mesh into a uniform latent space suitable for FNO application, and subsequently reversing this deformation. Pokle et al. (2022) propose FNO-DEQ, a Deep Equilibrium Model (DEQ (Bai et al., 2019)) variant with Fourier layers, to solve steady-state PDEs, showing improvements in accuracy and robustness to noise over baselines with four times as many parameters. Kohl et al. (2023) demonstrated that diffusion models are viable for turbulent flow simulation. Their results show that diffusion models outperform, in terms of long-term accuracy and stability, more efficient (and more commonly used) neural simulators. Recently, transformer-based architectures have gained prominence. Alkin et al. (2024) introduce the Universal Physics Transformer, a unified Eulerian-Lagrangian framework capable of handling large-scale simulations. Separately, McCabe et al. (2023) show that a single transformer pre-trained on multiple physics tasks can match or exceed task-specific baselines without additional fine-tuning.

These diverse architectures have demonstrated the potential of neural simulators across various scientific domains. Luz et al. (2020) performed experiments on a broad class of problems demonstrating improved convergence rates compared to highly efficient numerical methods. Pathak et al. (2020) found that the machine learning-assisted coarse-grid evolution resulted in corrected solution trajectories that were consistent with the solutions at a much higher resolution in space and time using highly-efficient spectral solvers. Stevens & Colonius (2020) proposed a method that outperforms a highly efficient numerical method in simulations where the numerical solution becomes overly diffused due to numerical viscosity.[1] Aurora (Bodnar et al., 2024), a foundation model for the Earth system, outperforms operational forecasts in predicting air quality, ocean waves, tropical cyclone tracks and high-resolution weather, all at orders of magnitude lower computational costs. Aurora generates predictions in approximately 0.6s per hour lead time on a single A100 GPU—this yields roughly a 100,000 times speed-up over CAMS. The fine-tuned model improves on all targets with an average magnitude of 54%. While these advances demonstrate the potential of neural simulators, explicit test-time control of the accuracy-cost trade-off remains largely unexplored.

---

[1]The results of Luz et al. (2020), Pathak et al. (2020), and Stevens & Colonius (2020) have been independently verified by McGreivy & Hakim (2024) and have been deemed "fair". The verification process ensured comparisons satisfied two critical criteria: (i) comparisons at equal accuracy or equal runtime, and (ii) comparison against efficient numerical methods appropriate for each specific PDE.

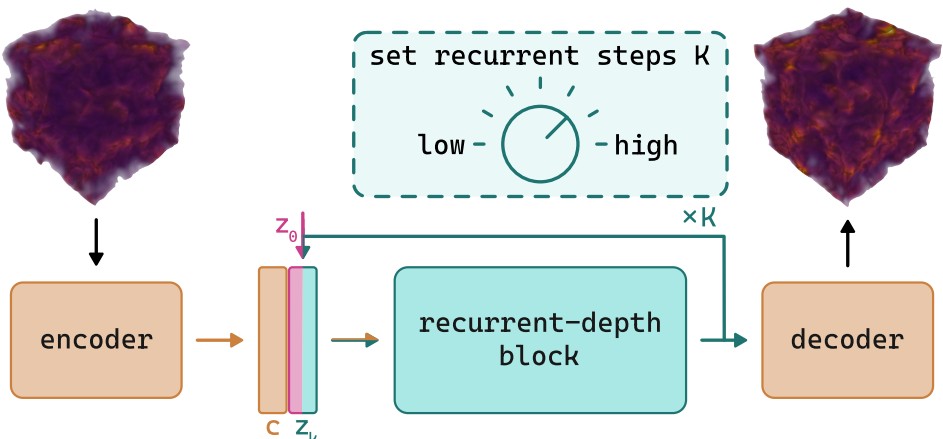

Figure 1: Schematic of the **Recurrent-Depth Simulator** (**RecurrSim**) framework. RecurrSim consists of three main components: an encoder, a recurrent-depth block, and a decoder. At test-time, the user is able to control the accuracy-cost trade-off by setting the number of recurrent iterations $K$.

## 3   RECURRENT-DEPTH SIMULATOR

**Overview.** The proposed **Recurrent-Depth Simulator** (**RecurrSim**) framework consists of three main components: an encoder, a recurrent-depth block, and a decoder (see Figure 1). The encoder transforms the input state $\mathbf{x}$ into a conditioning vector $\mathbf{c}$. An initial latent $\mathbf{z}_0$ is drawn from a fixed distribution $p(\mathbf{z})$ (which may be deterministic). For a user-chosen number of recurrent iterations $K$, the recurrent-depth block $\mathcal{R}(\cdot, \theta_{\mathcal{R}})$—conditioned on $\mathbf{c}$—iteratively updates the latent:

$$\mathbf{z}_k = \mathcal{R}\left([\mathbf{c}, \mathbf{z}_{k-1}], \theta_{\mathcal{R}}\right), \qquad k = 1, \dots, K.$$

After the final recurrent iteration, the decoder maps $\mathbf{z}_K$ to the predicted state $\hat{\mathbf{y}}$.

**Training.** RecurrSim is trained end-to-end. For each training sample, a number of recurrent iterations $K$ is drawn from a distribution $p(K)$, the recurrent-depth block is applied that many times, a supervised loss is evaluated, and gradients are back-propagated through the computation (see Algorithm 1). Sampling a wide range of recurrent iterations $K$ encourages the recurrent block to contract toward a fixed point.

A large number of recurrent iterations $K$ inflates memory because every intermediate activation must ordinarily be stored. To bound the memory footprint, we use truncated backpropagation-through-*depth* with a fixed *backpropagation window* $B$ (Williams & Peng, 1990). Gradients are propagated through, at most, the last $B$ recurrent iterations, while earlier iterations are treated as constants. This caps memory at $O(B)$ regardless of $K$ and has proved sufficient for optimization. Empirically, performance saturates at $B = 4$, with larger values yielding diminishing returns but significantly increasing memory usage (see Appendix D).

**Inference.** At test-time, the user is free to choose the number of recurrent iterations $K$ according to their desired accuracy and available computational resources (see Algorithm 2). Choosing a small $K$ generates fast, less-accurate simulations ideal for exploratory runs, or real-time control loops; whereas choosing a large $K$ value generates more-accurate, but slow, simulations suitable for critical applications or offline studies. Empirically, the first few recurrent steps make the largest adjustments to the latent vector $\mathbf{z}_k$; and subsequent steps contribute progressively smaller, yet still beneficial, adjustments. This behavior mirrors numerical methods, such as fixed-point and Newton methods, endowing RecurrSim with a strong inductive bias suitable for scientific computing tasks.

**Modularity.** RecurrSim is modular: each of, the encoder, recurrent-depth block, and decoder may be instantiated with the architecture primitive best suited to the problem—e.g., convolutional layers for Eulerian simulations or graph-convolutional layers for Lagrangian simulations—without altering training or inference algorithms. The entire framework remains a standard end-to-end, supervised model with no custom losses, schedulers, or tricks, so adoption is essentially plug-and-play.

---

**Algorithm 1** Recurrent-Depth Simulator Training

---

**Input:** training data $\mathbf{x}, \mathbf{y}$
**Output:** model parameters $\boldsymbol{\theta}_{\mathcal{E}}, \boldsymbol{\theta}_{\mathcal{R}}, \boldsymbol{\theta}_{\mathcal{D}}$
**repeat**
    **for** $i \in \mathcal{B}$ **do**                       ▷ for every training example index in batch
        $\mathbf{c} \leftarrow \mathcal{E}(\mathbf{x}_i, \boldsymbol{\theta}_{\mathcal{E}})$                 ▷ compute conditioning vector
        $\mathbf{z}_0 \sim p(\mathbf{z})$                  ▷ sample initial latent representation
        $K \sim p(K)$                  ▷ sample number of recurrent iterations
        **for** $k = 1$ to $K$ **do**               ▷ unroll $K$ recurrent iterations
            $\mathbf{z}'_{k-1} \leftarrow [\mathbf{c}, \mathbf{z}_{k-1}]$     ▷ concatenate conditioning and latent representation
            $\mathbf{z}_k \leftarrow \mathcal{R}(\mathbf{z}'_{k-1}, \boldsymbol{\theta}_{\mathcal{R}})$           ▷ apply recurrent block
        **end for**
        $\hat{\mathbf{y}}_i \leftarrow \mathcal{D}(\mathbf{z}_K, \boldsymbol{\theta}_{\mathcal{D}})$             ▷ decode latent representation
        $l_i \leftarrow \|\mathbf{y}_i - \hat{\mathbf{y}}_i\|$               ▷ compute individual loss
    **end for**
    accumulate losses for batch and take gradient step
**until** converged

---

**Algorithm 2** Recurrent-Depth Simulator Inference

---

**Input:** input state $\mathbf{x}$, number of recurrent iterations $K$, model parameters $\boldsymbol{\theta}_{\mathcal{E}}, \boldsymbol{\theta}_{\mathcal{R}}, \boldsymbol{\theta}_{\mathcal{D}}$
**Output:** output state $\mathbf{y}$
$\mathbf{c} \leftarrow \mathcal{E}(\mathbf{x}, \boldsymbol{\theta}_{\mathcal{E}})$                  ▷ compute conditioning vector
$\mathbf{z}_0 \sim p(\mathbf{z})$                   ▷ sample initial latent representation
**for** $k = 1$ to $K$ **do**               ▷ unroll $K$ recurrent iterations
    $\mathbf{z}'_{k-1} \leftarrow [\mathbf{c}, \mathbf{z}_{k-1}]$       ▷ concatenate conditioning and latent representation
    $\mathbf{z}_k \leftarrow \mathcal{R}(\mathbf{z}'_{k-1}, \boldsymbol{\theta}_{\mathcal{R}})$             ▷ apply recurrent block
**end for**
$\mathbf{y} \leftarrow \mathcal{D}(\mathbf{z}_K, \boldsymbol{\theta}_{\mathcal{D}})$           ▷ decode latent representation to predicted state

---

**Initial Latent Distribution.** The initial latent vector $\mathbf{z}_0$ is drawn from a fixed distribution $p(\mathbf{z})$. Common choices include degenerate distributions (e.g., zeros or average of target fields), standard normal distributions, or heavy-tailed alternatives like Student-$t$ priors (Pandey et al., 2025). Empirically, this choice primarily affects early iterations, with minimal impact later as the latent converges toward the fixed point. We use the standard normal distribution $\mathcal{N}(\mathbf{0}, \mathbf{I})$ for consistency with DEQ and diffusion models.

**Recurrent Iteration Distribution.** The number of recurrent iterations $K$ is drawn from a Poisson log-normal distribution:

$$v \sim \mathcal{N}\left(\log \bar{K} - \frac{1}{2}\sigma^2, \sigma\right),$$
$$K \sim \text{Poisson}\left(e^v\right) + 1,$$

where $\bar{K} + 1$ is the desired mean (Geiping et al., 2025). This distribution exposes the model to a broad spectrum of compute budgets during training: it is positively skewed with most draws landing near $\bar{K}$, but occasional very small and very large values are sampled, encouraging the recurrent-block to remain stable across both shallow and deep rollouts. Empirically, performance saturates around $\bar{K} = 32$ ($\sigma = 0.5$), with larger values providing diminishing returns (see Appendix E).

**Merging Conditioning and Latent Vectors.** At each recurrent iteration, the conditioning vector $\mathbf{c}$ is merged with the current latent vector $\mathbf{z_k}$. The most straightforward scheme is plain addition: $\mathbf{z}'_k = \mathbf{c} + \mathbf{z}_k$. A slightly richer variant introduces learnable scalar weights: $\mathbf{z}'_k = \alpha\mathbf{c} + \beta\mathbf{z}_k$; the weights can be made element-wise: $\mathbf{z}'_k = \boldsymbol{\alpha}\odot\mathbf{c} + \boldsymbol{\beta}\odot\mathbf{z}_k$. Alternatives include point-wise projection, or concatenating and passing the result through a width-halving layer. Empirically, addition with element-wise weights offers the best balance between parameter efficiency and performance (see Appendix H).

## 4 RESULTS

Complete experimental details, including hardware specifications, data acquisition, data generation, preprocessing pipelines, training pipelines, and architectural configurations, are provided in Appendix A-C. We conduct experiments across diverse datasets including Burgers Equation, Korteweg-De Vries Equation, Kuramoto-Sivashinsky Equation, Compressible Navier-Stokes Equations, Active Matter, and ShapeNet-Car; and across multiple architectural backbones including Fourier Neural Operator (RecurrFNO), Vision Transformer (RecurrViT), and Universal Physics Transformer (RecurrUPT).

### 4.1 EQUATIONS

**Burgers Equation.**   The Burgers equation is a second-order nonlinear partial differential equation derived to model convective steepening and diffusive smoothing. Its one-dimensional variant can be expressed as:
$$u_t + uu_x = \nu u_{xx}.$$
Here, $\nu$ plays the role of kinematic viscosity. Setting $\nu = 0$ yields the inviscid form $u_t + uu_x = 0$, whose solutions develop finite-time shock discontinuities; the viscous term $\nu u_{xx}$ regularises these shocks but introduces extremely thin internal layers that remain numerically stiff. Machine learning methods must learn to represent sharp gradients, moving shocks and the delicate interplay between nonlinearity and diffusion.

**Korteweg-De Vries Equation.**   The Korteweg-De Vries (KdV) is a third-order nonlinear partial differential equation derived to model weakly nonlinear, weakly dispersive unidirectional waves. Its one-dimensional variant can be expressed as:
$$u_t + \alpha uu_x + u_{xxx} = 0.$$
Here, $\alpha$ (often set to $\pm 1$ or $\pm 6$) controls nonlinear steepening while the third-order derivative $u_{xxx}$ introduces dispersion. The exact balance of these effects produces solitary-wave solutions (solitons) that preserve their shape and speed and undergo only phase shifts upon interaction –small amounts of artificial dissipation can destroy these very structures making KdV an ideal candidate for evaluating whether machine learning methods can maintain accuracy, stability and conservation over long horizons.

**Kuramoto-Sivashinsky Equation.**   The Kuramoto-Sivashinsky (KS) equation is a fourth-order nonlinear partial differential equation derived to model diffusive-thermal instabilities in laminar flame fronts. Its one-dimensional variant can be expressed as:
$$u_t + u_{xx} + u_{xxxx} + uu_x = 0.$$
Here, the fourth-order derivative $u_{xxxx}$ and the nonlinear term $uu_x$ contribute to complex and chaotic behavior which present a challenge for traditional numerical solvers. The challenges and the wide applicability of the KS equation make it an ideal candidate for evaluating machine learning methods.

**Compressible Navier-Stokes Equations.**   The three-dimensional Compressible Navier-Stokes (CNS) equations model complex phenomena such as shock wave formation and propagation. They are widely used across various engineering and physics applications, including aircraft wing aerodynamics and the formation of interstellar gases. The equations can be expressed as:
$$\partial_t \rho + \nabla \cdot (\rho \mathbf{v}) = 0, \quad \rho(\partial_t \mathbf{v} + \mathbf{v} \cdot \nabla \mathbf{v}) = -\nabla p + \eta \Delta \mathbf{v} + (\zeta + \eta/3)\nabla(\nabla \cdot \mathbf{v}),$$
$$\partial_t(\epsilon + \rho \mathbf{v}^2/2) + \nabla \cdot \left[ (p + \epsilon + \rho \mathbf{v}^2/2)\mathbf{v} - \mathbf{v} \cdot \sigma' \right] = 0,$$
where $\rho$ is the mass density, $\mathbf{v}$ is the fluid velocity, $p$ is the pressure, and $\epsilon$ is the internal energy determined by the equation of state. The term $\sigma'$ denotes the viscous stress tensor, while $\eta$ and $\zeta$ represent the shear and bulk viscosities, respectively. In this case, using a classical numerical solver to approximate the fluid flow is particularly challenging due to strict stability constraints, high computational cost, and the need for accurate yet robust numerical schemes that handle shocks, dissipation, and grid adaptivity in large-scale domains. Even though machine learning can overcome several of the challenges posed by traditional solvers, training a neural simulator on three-dimensional data comes with considerable engineering complexity. The primary limitation arises from storing the activations during training, increasing the memory requirement compared to smaller dimensions problems.

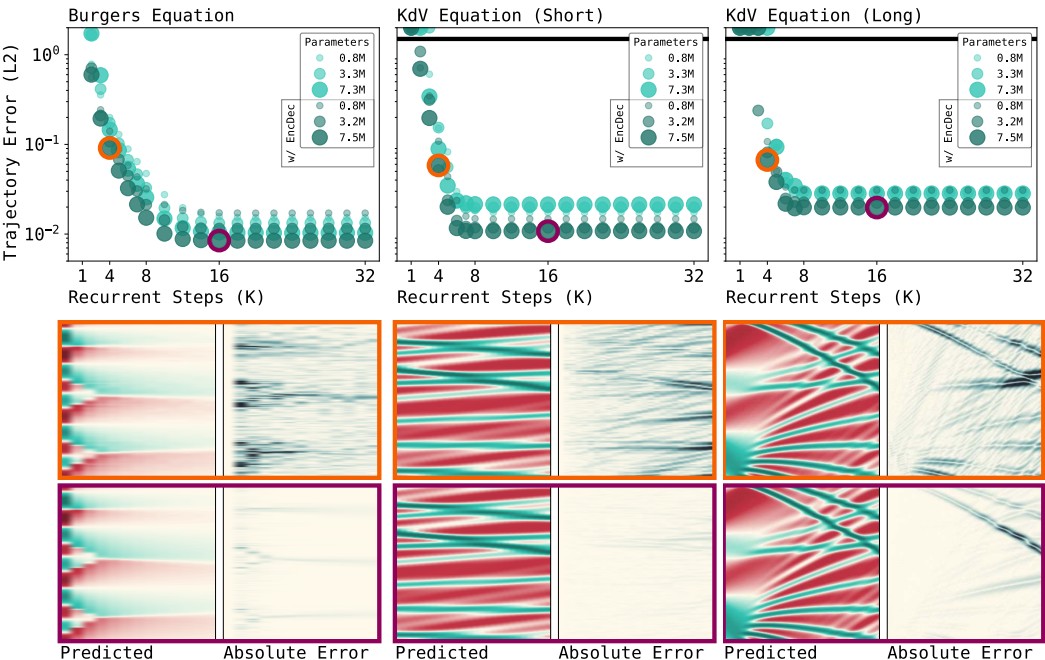

Figure 2: **Top:** Trajectory Error (L2) versus Recurrent Steps (K) for the Burgers (left), short-horizon KdV (middle), and long-horizon KdV (right). **Bottom:** Trajectories at $K = 4$ (orange) and $K = 16$ (purple) (highlighted above). Increasing $K$ sharpens shocks in Burgers and aligns soliton crests in KdV, illustrating how recurrent depth controls the accuracy–cost trade-off. See Figure 13 for detailed visualization of **Bottom** with ground truth and colorbars.

## 4.2 EXPERIMENT: ACCURACY-COST TRADE-OFF

Commonly used neural simulators are trained for a single accuracy-cost setting: once the model is trained, every forward pass delivers the same expected accuracy and incurs the same cost. RecurrSim, on the other hand, has a tunable knob for controlling the accuracy-cost setting (the number of recurrent iterations $K$). The purpose of this experiment is to empirically demonstrate whether rolling out the trajectory across values of recurrent iterations $K$ is viable.

**Experimental Setup.** We conduct experiments on three datasets: Burgers, short-horizon KdV, and long-horizon KdV. Two instantiations of RecurrSim are benchmarked. The first variant (RecurrFNO `wo/ EncDec`) lifts the input with a point-wise operation, recursively applies a recurrent-depth block with a single Fourier layer, and projects back to physical space; the second variant (RecurrFNO `w/ EncDec`) inserts an additional Fourier layer in, both, the encoder and decoder. For each variant, we target three parameter budgets ($\sim 1.0M, 3.5M, 7.5M$), yielding six models per dataset. We use $\bar{K} = 32$ and $B = 4$. After convergence, we generate trajectories for every $K \in \{1, \ldots, 32\}$ and measure the trajectory error. All experiments are repeated with three seeds and averaged.

**Results.** Across all three datasets, both variants show the same qualitative accuracy-cost curve (Figure 2), but RecurrFNO `w/ EncDec` achieves consistently lower trajectory error. As $K$ increases, the trajectory error falls steadily and plateaus around $K = 16$ for Burgers and $K = 8$ for both, short- and long-horizon KdV ; further steps neither help nor harm. For each dataset, we plot low-compute ($K = 4$) and high-compute ($K = 16$) trajectories. In Burgers, the two settings reproduce the same shock patterns, with the low-compute run showing slightly larger absolute error around the fronts. In both KdV datasets, the low-compute run already recovers the full soliton train; the absolute error is almost entirely a small amplitude and/or phase offset, visible as narrow streaks along the soliton trajectories. Increasing to $K = 16$ sharpens the shocks and aligns the soliton crests. These results demonstrate that RecurrFNO delivers physically faithful simulations over a range of accuracy-cost settings. Extended results are presented in Appendix I.

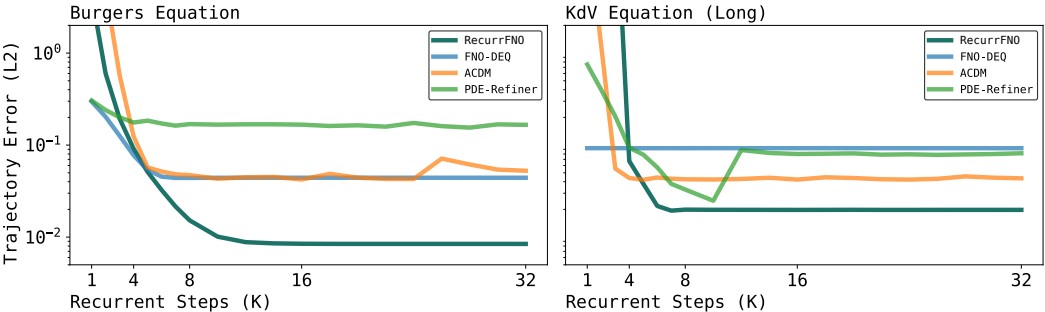

Figure 3: Trajectory Error (L2) versus Recurrent Steps (K) for the Burgers (left), and long-horizon KdV (right). Curves compare RecurrFNO (teal), FNO-DEQ (blue) (Marwah et al., 2023), ACDM (orange) (Kohl et al., 2023), and PDE-Refiner (green) (Lippe et al., 2023). Across both tasks, RecurrFNO achieves the best accuracy-cost curve and reaches the lowest plateau.

## 4.3 EXPERIMENT: ALTERNATIVES

There are a few recent neural simulators that have test-time controllable knobs. FNO-DEQ is a Deep Equilibrium Model with Fourier layers whose runtime is set by a maximum number of iterations or a minimum update. ACDM—an autoregressive conditional diffusion model—is able to adjust the prediction quality by varying the number and schedule of denoising steps. PDE-Refiner applies the same diffusion principle in a direct prediction and refinement process. In this experiment, we benchmark RecurrFNO against the three alternatives under identical data and training setups.

**Experimental Setup.** We conduct experiments on three datasets: Burgers, long-horizon KdV, and long-horizon Kuramoto-Sivashinsky. For RecurrSim, we carry over the best variant from the previous set of experiments: point-wise lift + Fourier layer encoder, a recurrent-depth block with one Fourier layer, Fourier layer with point-wise projection decoder—configured with $\sim 7.5\text{M}$ parameters, $\bar{K} = 32$, and $B = 4$ steps. FNO-DEQ follows the setup of Pokle et al. (2022), with its width scaled to match a parameter count of $\sim 7.5\text{M}$. ACDM and PDE-Refiner use a modern UNet backbone from their original implementations (Kohl et al., 2023; Lippe et al., 2023). In early tests, both diffusion-based models proved parameter-inefficient and could not rollout beyond a few steps, so we train them with $\sim 15\text{M}$ parameters for Burgers and KdV, and $\sim 50\text{M}$ parameters for KS (the scale used by Lippe et al. (2023)). After convergence, we generate trajectories for every $K \in \{1, \ldots, 32\}$ (where $K$ is equal to the recurrent steps for RecurrSim, iterations for FNO-DEQ, and denoising steps for ACDM and PDE-Refiner). On Burgers and KdV, we measure and report the trajectory error. Since the KS equation produces chaotic behavior, we measure the average and worst-case correlation horizon over a sweep of 30 thresholds ($\alpha = 0.7$-$0.99$ in increments of $0.01$).

**Results.** On Burgers, FNO-DEQ, ACDM, and PDE-Refiner all plateau by $K \approx 4$ (see Figure 3 (left)); PDE-Refiner gains practically nothing beyond its second refinement step. RecurrFNO, by contrast, continues to improve until $K \approx 16$, while using half the parameters of the diffusion-based models. On KdV, FNO-DEQ exhibits the convergence limitation reported by Sittoni & Tudisco (2024)—the latent representation oscillates around, rather than converges to, the fixed point—so additional iterations provide no improvement. The ten-fold larger training dataset helps the diffusion-based models, however, once again, ACDM plateaus near $K \approx 4$. PDE-Refiner improves up to $K = 11$ before degrading because larger $K$ values are out-of-distribution. RecurrFNO delivers the best accuracy-cost curves and lowest trajectory errors. On KS (see Appendix I), where the diffusion-based models have 7-fold the amount of parameters as RecurrFNO, ACDM plateaus early, and PDE-Refiner shows erratic worst-case correlation horizons. Taken together, RecurrFNO consistently outperforms alternatives while using fewer parameters.

| Model | Params | Training Memory | Training Epochs | Training GFLOPs | MSE $\times 10^{-2}$ Density | MSE $\times 10^{-2}$ Pressure | MSE $\times 10^{-2}$ Velocity |
|---|---|---|---|---|---|---|---|
| FNO | 0.5B | 38 GB | 100 | $1 \times 10^7$ | $9.60 \pm 0.03$ | $9.59 \pm 0.03$ | $9.55 \pm 0.03$ |
| FNO | 1.0B | 57 GB | 100 | $2 \times 10^7$ | $7.83 \pm 0.02$ | $7.79 \pm 0.02$ | $7.82 \pm 0.03$ |
| FNO | 1.6B | 73 GB | 100 | $3 \times 10^7$ | $7.61 \pm 0.01$ | $7.59 \pm 0.02$ | $7.62 \pm 0.03$ |
| RecurrFNO | 0.8B | 64 GB | 82 | $3 \times 10^7$ | $\textbf{7.57} \pm \textbf{0.04}$ | $\textbf{7.51} \pm \textbf{0.01}$ | $\textbf{7.53} \pm \textbf{0.03}$ |
| RecurrFNO | 0.8B | 64 GB | 100 | $5 \times 10^7$ | $\textbf{7.37} \pm \textbf{0.03}$ | $\textbf{7.33} \pm \textbf{0.01}$ | $\textbf{7.36} \pm \textbf{0.03}$ |

Table 1: Comparison between FNO and RecurrFNO on 3D Compressible Navier-Stokes Equations.

| Model | Params | MSE $\times 10^{-2}$ Steps 0:3 | MSE $\times 10^{-2}$ Steps 0:6 | MSE $\times 10^{-2}$ Steps 0:12 |
|---|---|---|---|---|
| ViT | 130M | 2.91 | 12.41 | 43.16 |
| RecurrViT | 75M | **0.54** | **1.31** | **5.68** |

Table 2: Comparison between ViT and RecurrViT on Active Matter.

| Model | Resolution | Params | MSE $\times 10^{-2}$ Original Work | MSE $\times 10^{-2}$ Our Work |
|---|---|---|---|---|
| UPT | $32^3$ | 164M | 2.35 | 2.31 |
| RecurrUPT | $32^3$ | 92M | N/A | **2.19** |

Table 3: Comparison between UPT and RecurrUPT on ShapeNet-Car.

## 4.4 Experiment: High-Dimensional Simulations and Transformer Variants

Many real-world scientific simulations involve high-dimensional problems where memory-intensive approaches like ADCM and PDE-Refiner become computationally prohibitive. Additionally, the generalizability of RecurrSim across different architectural primitives and problem domains remains to be demonstrated. We address these challenges through three targeted experiments: (1) evaluating memory efficiency on high-dimensional 3D compressible Navier-Stokes equations (262k grid points) where traditional deep networks face memory constraints, (2) demonstrating architectural flexibility by adapting Kohl et al. (2023)'s vision transformers for active matter simulations (Ohana et al., 2025), and (3) validating the framework's drop-in compatibility by recreating UPT's (Alkin et al., 2024) ShapeCar-Net experiments, only adapting their approximator module by decreasing the depth and wrapping it in a recurrent-block—this significantly lowers the number of trainable parameters.

**Results.** RecurrSim variants consistently achieve superior accuracy with dramatically reduced computational requirements across all domains (Table 1, Table 2, Table 3). On 3D compressible Navier-Stokes equations, RecurrFNO with $\bar{K} = 8$ outperforms all FNO baselines, including a 1.6B parameter model, while requiring less memory (64GB vs 73GB). When training epochs are matched ($\bar{K} = 16$), RecurrFNO achieves MSE improvements on density, pressure, and velocity compared to the strongest FNO baseline. On Active Matter, RecurrViT with 75M parameters (58% of ViT's 130M) reduces error accumulation by 83% at 3 timesteps and maintains 87% lower error at 12 timesteps. On ShapeNet-Car, RecurrUPT with 92M parameters (56% of UPT's 164M) achieves better performance to the original work, demonstrating perfect drop-in compatibility. These results establish that RecurrSim provides a universal framework for test-time accuracy-cost control: users can deploy a single trained model across diverse computational budgets simply by adjusting the number of recurrent iterations.

## 5 Conclusions and Future Work

We introduce the Recurrent-Depth Simulator (RecurrSim), an architecture-agnostic framework enabling explicit test-time control over accuracy-cost trade-offs in neural simulators. By adjusting the number of recurrent iterations K at inference, users can deploy a single trained model across diverse computational budgets—from fast exploratory runs to high-accuracy critical simulations. RecurrSim demonstrates physically faithful behavior across fluid dynamics benchmarks and consistently outperforms existing adaptive methods (FNO-DEQ, ACDM, PDE-Refiner) while using fewer parameters. On high-dimensional problems, RecurrFNO with 0.8B parameters outperforms 1.6B FNO baselines using 13.5% less memory, while RecurrViT and RecurrUPT show similar advantages across transformer architectures. This plug-and-play framework fundamentally improves neural simulator utility and opens new research directions in adaptive scientific computing.

**Ethics Statement.** We adhere to the ICLR Code of Ethics. Our work does not involve human subjects, create potentially harmful insights, raise discrimination/bias/fairness concerns, or pose privacy and security risks. Our work uses data generated from numerical solvers (which we make publicly available) and publicly available datasets. The aim of this work is to advance AI for Science. Although related methods may be deployed in critical applications, we do not deploy in such contexts. We have taken care to report results honestly, acknowledge limitations, and follow best practices for research integrity.

**Reproducibility Statement.** We have made significant effort to ensure reproducibility. We use clear and consistent terminology throughout the work. Section 3 describes our framework in detail, explaining each design choice and linking to ablation studies in the Appendix. Section 4 describes the motivation and practical implementation of each experiment, with additional results in the Appendix. The Appendices provide complete descriptions of hardware specifications (Appendix A), data generation and processing (Appendix B), hyperparameters and optimization details (Appendix C, and pseudocode (Appendix J). We also provide source code for all experiments. Together, these resources enable independent researchers to reproduce and verify our findings.

**Acknowledgments.** The work of FT is partially funded by the PRIN-MUR project MOLE code: 2022ZK5ME7 and by the PRIN-PNRR project FIN4GEO within the European Union's Next Generation EU framework, Mission 4, Component 2, CUP P2022BNB97

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

# A HARDWARE DETAILS

For the one-dimensional Burgers, Korteweg-De Vries, and Kuramoto-Sivashinsky equations, we generated the data using an AMD 7950X processor (16 cores/32 threads). Each example trajectory in the Burgers equation and Korteweg-De Vries equation datasets took approximately 10 and 20 minutes to generate, respectively. The entire datasets, with 600 examples (500 training examples and 100 testing examples), took approximately 6000 and 12000 minutes to generate, respectively. Each training example in the Kuramoto-Sivashinsky equation dataset took approximately 15 minutes to generate. The testing examples were twice as long, and took approximately 30 minutes to generate. The entire dataset, with 500 training examples and 100 testing examples, took approximately 10500 minutes to generate. All together, the three one-dimensional datasets took approximately 28500 minutes (475 hours) to generate.

All one-dimensional models were trained on a single NVIDIA A100 (40GB) GPU per run, with average training times ranging from 15-300 minutes per model—smaller models on the Burgers dataset took 15 minutes, whereas larger models trained on Korteweg-De Vries or Kuramoto-Sivashinsky datasets, which contained 10 times longer trajectories, took closer to 300 minutes. We trained approximately 1000 models for exploratory experiments (e.g., tuning hyperparameters, evaluating alternative architectures) and final experiments, and estimate a total of 1000 NVIDIA A100 (40GB) GPU hours.

The three-dimensional models were much larger. Under our experimental setup, only the smallest Fourier Neural Operator with two layers managed to fit on a single NVIDIA A100 (40GB) GPU. This model did not perform well (approximately 25-30% higher MSE compared to its six layer variant). So all three-dimensional experiments were trained on a single NVIDIA A100 (80GB) GPU. On average, each training run took 1200-1500 minutes to complete. We trained approximately 10 models for exploratory experiments and final experiments, and estimate a total of 225 NVIDIA A100 (80GB) GPU hours.

# B DATA DETAILS

For the one-dimensional Burgers and Korteweg-De Vries equations, we set $T = 10$ and $T = 100$, respectively (for both training and testing datasets). For the one-dimensional Kuramoto-Sivashinsky equation, we set $T = 100$ for the training dataset and $T = 200$ for the training dataset. For all three equations, we set $\Delta t = 0.2$. The spatial domain was set to $\mathcal{X} = [0, 2\pi]$ for the Burgers equation with $\Delta x = 2\pi/8192$, $\mathcal{X} = [0, 128]$ for the Korteweg-De Vries equation with $\Delta x = 128/1024$, and $\mathcal{X} = [0, 64]$ for the Kuramoto-Sivashinsky equation with $\Delta x = 64/4096$. For each equation, the spatial step $\Delta x$ was chosen to be as small as possible while maintaining trajectory generation under a pre-specified computational budget. All three domains had periodic boundaries. The initial conditions were sampled from a distribution over the truncated Fourier series with random coefficients $A_k \sim U(A_l, A_r)$, $l_k \sim \{l_a, l_b, l_c, l_d\}$, and $\phi_k \sim (\phi_l, \phi_r)$:

$$u_0(x) = \sum_{k=1}^{10} A_k \sin\left(\frac{2\pi l_k x}{L} + \phi_k\right),$$

where $L$ is the length of the spatial domain. Each trajectory was generated using the method of lines with the spatial derivatives computed using the pseudo-spectral method. For each equation, we selected a time-stepping method that balances accuracy and cost: RK23 for the Burgers equation, RK45 for the Korteweg-De Vries equation, and LSODA for the Kuramoto-Sivashinsky equation. See Table 4 for details.

| Equation | Train $T$ | Test $T$ | $\Delta t$ | $\mathcal{X}$ | $\Delta x$ | $\{A_l, A_r\}$ | $\{l_a, l_b, l_c, l_d\}$ | $\{\phi_l, \phi_r\}$ | Time-Stepping |
|---|---|---|---|---|---|---|---|---|---|
| Burgers | 10 | 10 | 0.2 | $[0, 2\pi]$ | $2\pi/8192$ | $\{-0.5, 0.5\}$ | $\{3, 4, 5, 6\}$ | $\{0, 2\pi\}$ | RK23 |
| KdV | 100 | 100 | 0.2 | $[0, 128]$ | $128/1024$ | $\{-0.5, 0.5\}$ | $\{1, 2, 3, -\}$ | $\{0, 2\pi\}$ | RK45 |
| KS | 100 | 200 | 0.2 | $[0, 64]$ | $64/4096$ | $\{-0.5, 0.5\}$ | $\{1, 2, 3, -\}$ | $\{0, 2\pi\}$ | LSODA |

Table 4: Data generation settings.

We construct two additional datasets, short-horizon Korteweg-De Vries and short-horizon Kuratmoto-Sivashinsky, by considering the first 400 time steps to be part of a *warmup phase* and subsequently discarding them. See Table Table 5 for details.

| Equation | Warm-Up Steps | Train $T$ | Test $T$ |
|---|---|---|---|
| Short-Horizon KdV | 400 | 20 | 20 |
| Short-Horizon KS | 400 | 20 | 120 |
| Long-Horizon KdV | 0 | 100 | 100 |
| Long-Horizon KS | 0 | 100 | 200 |

Table 5: Short-horizon and long-horizon settings.

For each of the one-dimensional equations, we generate 500 training trajectories and 100 testing trajectories. The data was initially generated using double-precision floating-point format (`float64`) and then converted into single-precision floating-point formation (`float32`) for our experiments.

THREE-DIMENSIONAL COMPRESSIBLE NAVIER-STOKES DATASET

We use the three-dimensional compressible Navier-Stokes turbulence dataset provided by Takamoto et al. (2022). This dataset consists of 600 trajectories, each containing 21 time steps, with 90% of the trajectories used for training and the remaining 10% reserved for testing. The turbulence initial condition considers turbulent velocity with uniform mass density and pressure. The initial velocity is defined as

$$\mathbf{v}(\mathbf{x}, t = 0) = \sum_{i=1}^{4} A_i \sin(\mathbf{k}_i \cdot \mathbf{x} + \phi_i),$$

where the amplitude coefficients are

$$A_i = \frac{\bar{v}}{|\mathbf{k}_i|^2},$$

and the characteristic velocity $\bar{v} = c_s M$ is determined by the Mach number $M$ and the speed of sound

$$c_s = \sqrt{\frac{\Gamma p}{\rho}}.$$

To reduce compressibility effects, the compressible component of the velocity field is removed using a Helmholtz decomposition in Fourier space, resulting in a divergence-free velocity field that preserves turbulent structures while minimizing artificial acoustic modes.

The flow parameters are set to

$$(\eta, \zeta, M) = \left(10^{-2}, 10^{-2}, 1.0\right),$$

where $\eta$ and $\zeta$ are the shear and bulk viscosity coefficients, respectively, and $M$ is the initial Mach number.

The data are simulated using a second-order accurate HLLC Toro et al. (1994) scheme for the inviscid terms, the MUSCL Van Leer (1997) method for spatial reconstruction, and a central difference scheme for the viscous terms.

Each time step is composed by five channels: the three velocity components, pressure, and density, and each time steps is represented on a $64^3$ grid, resulting in $5 \times 64^3 = 5 \times 262,144 \approx 1.31 \times 10^6$ data points per step. The whole dataset size is 62 GB, indeed, due to memory constraints, training is performed by loading sub-batches of 32 samples directly from the hard disk where the dataset was stored. While this approach slows down training, it is necessary given the large dataset size.

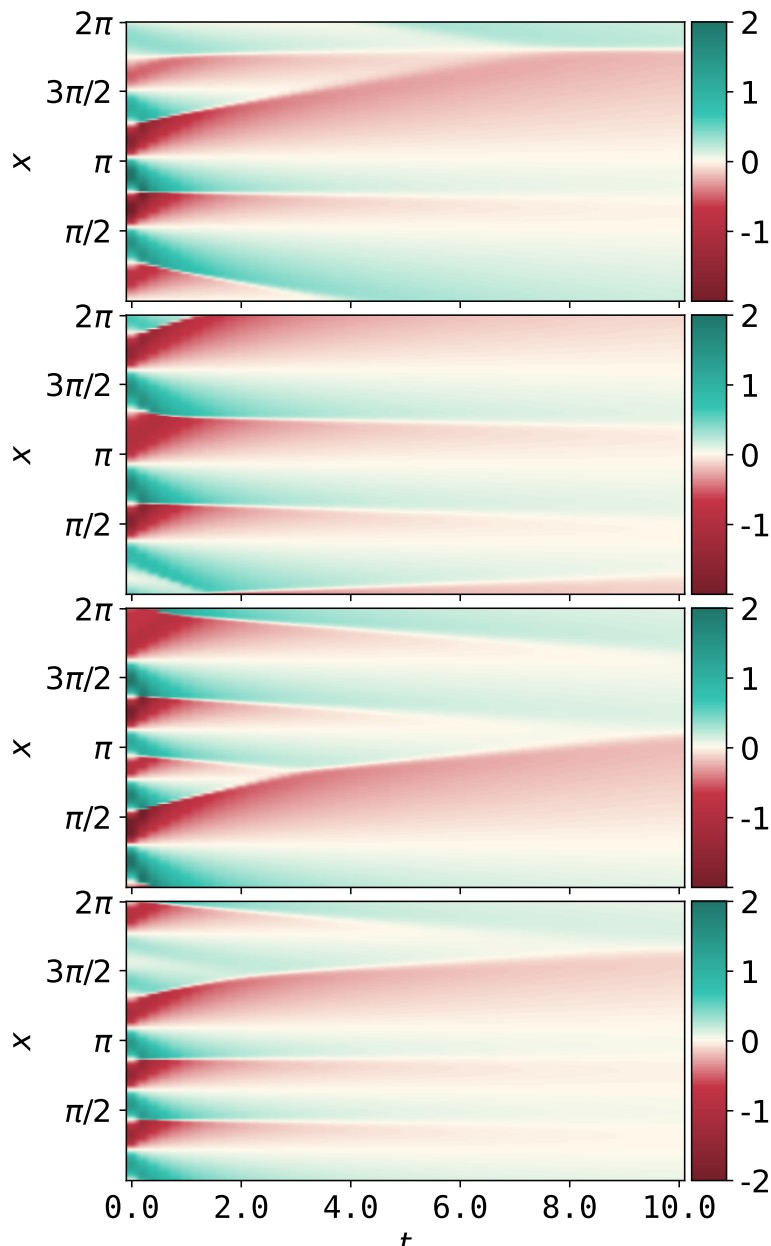

Figure 4: Example trajectories from the Burgers dataset. Train and test datasets share the same $T$.

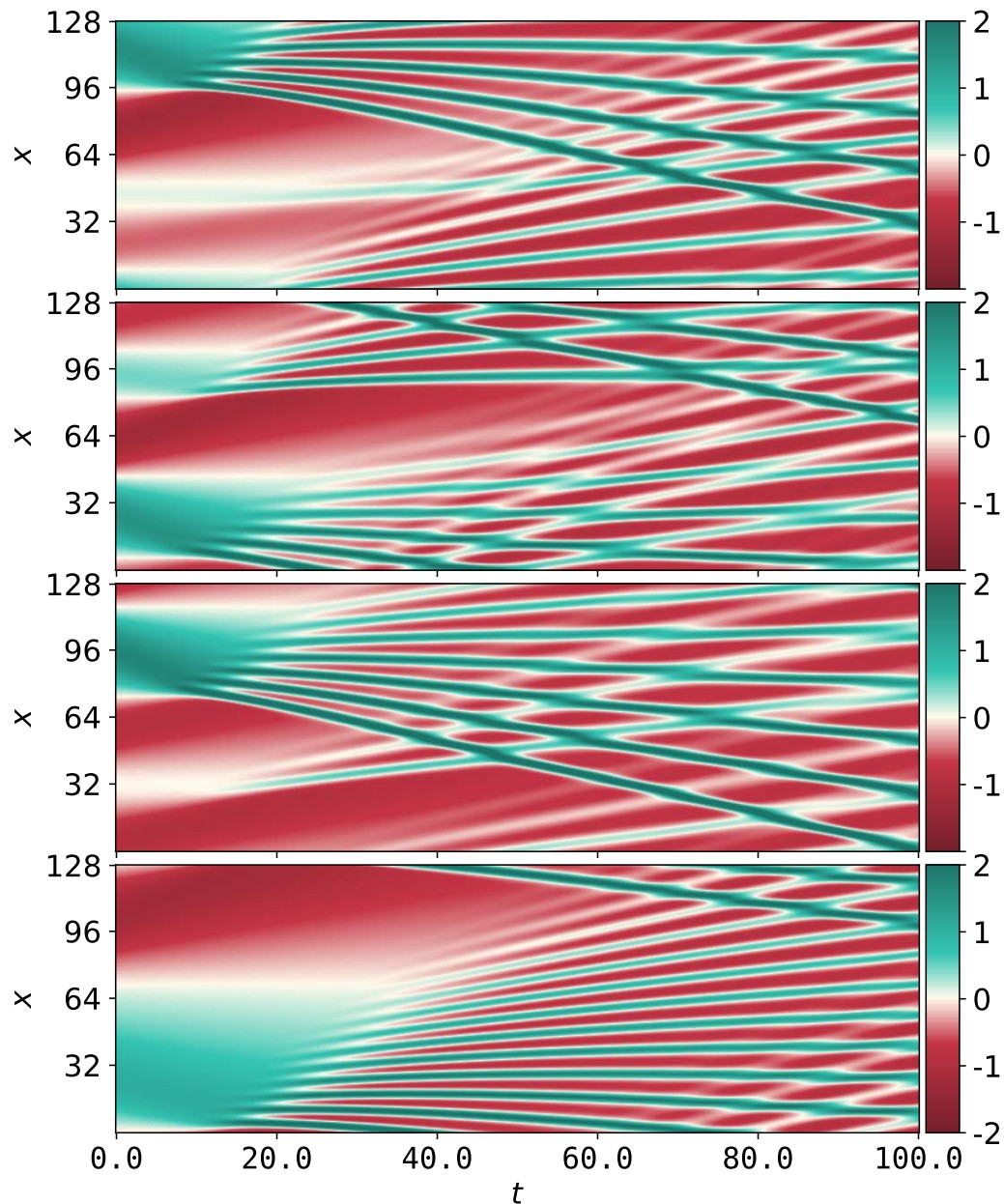

Figure 5: Example trajectories from the Korteweg-de Vries dataset. Train and test datasets share the same $T$.

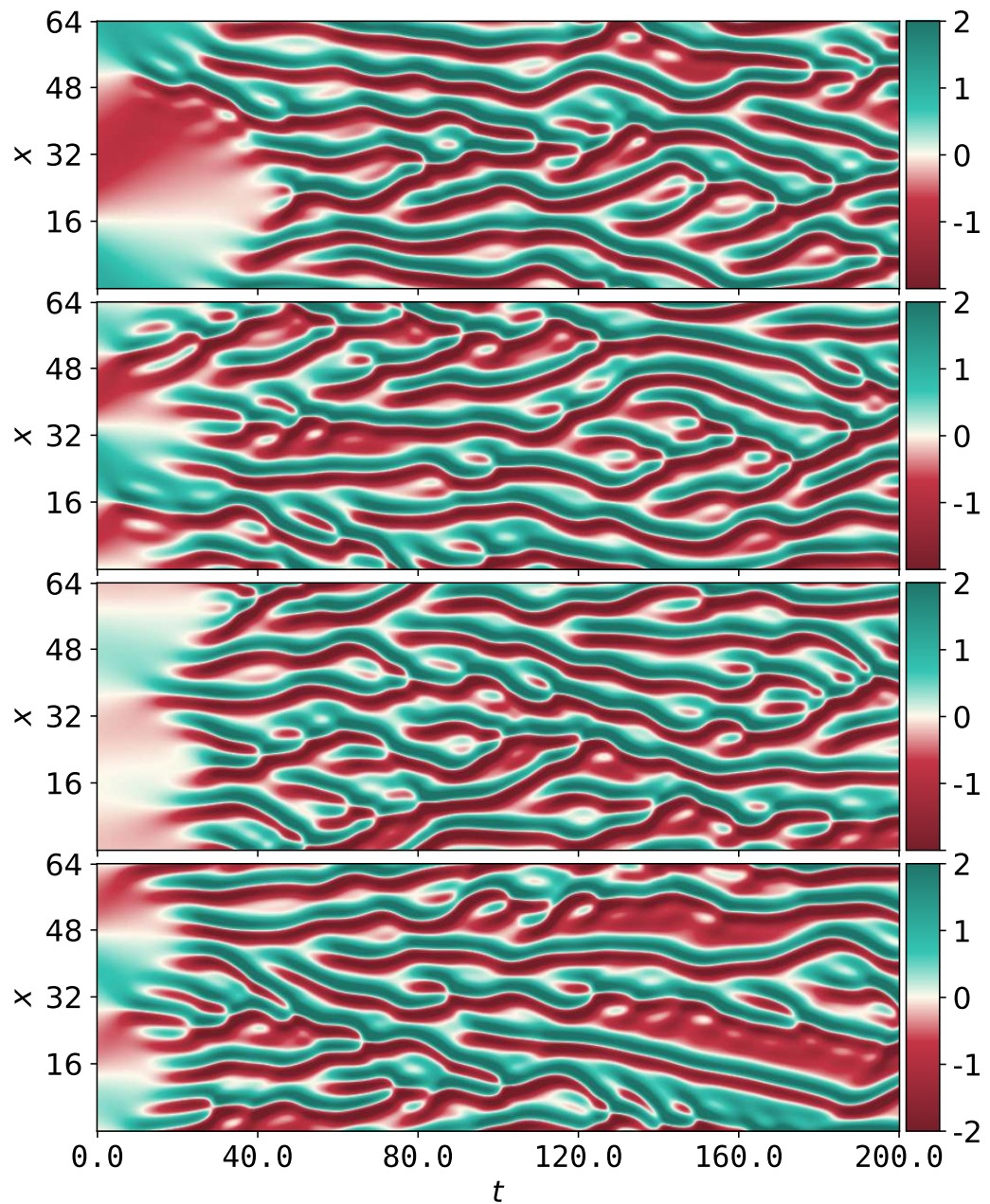

Figure 6: Example trajectories from the Kuramoto-Sivashinsky testing dataset. The training dataset has $T = 100$, whereas the testing dataset has $T = 200$.

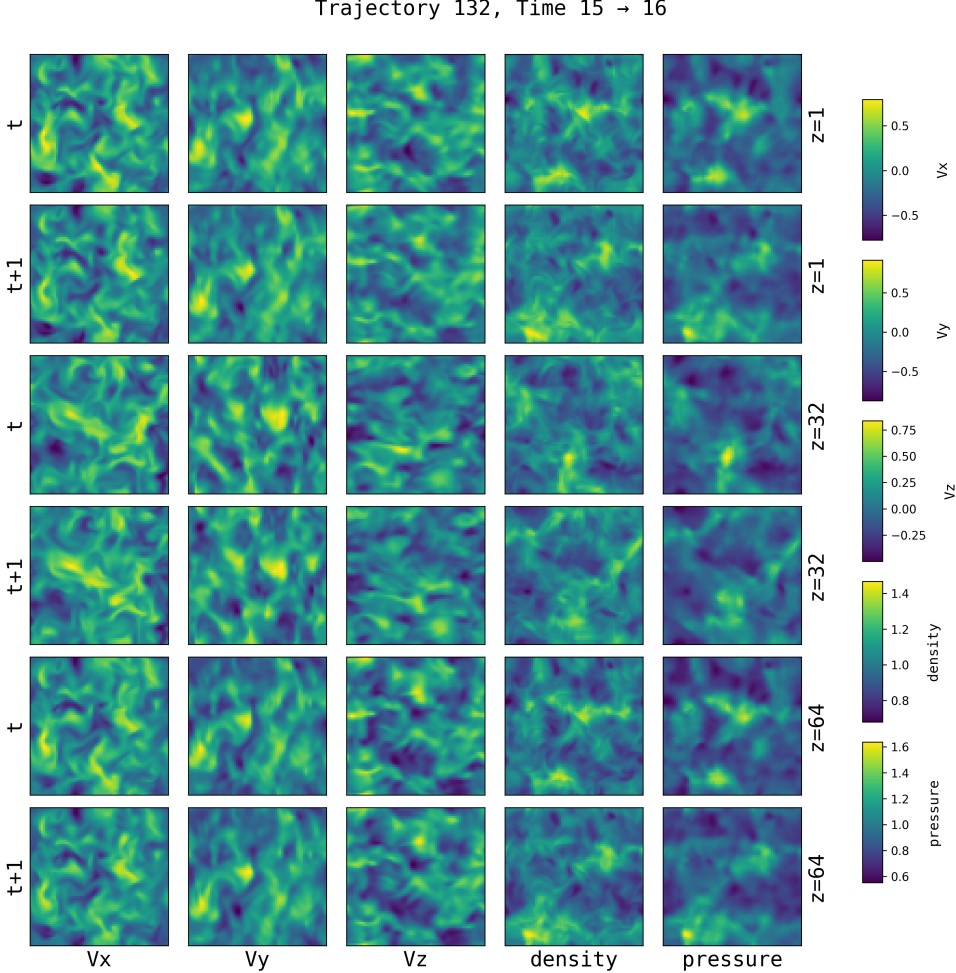

Figure 7: Example of two consecutive time steps from a three-dimensional Navier-Stokes simulation, visualized across different z-slices for all physical fields.

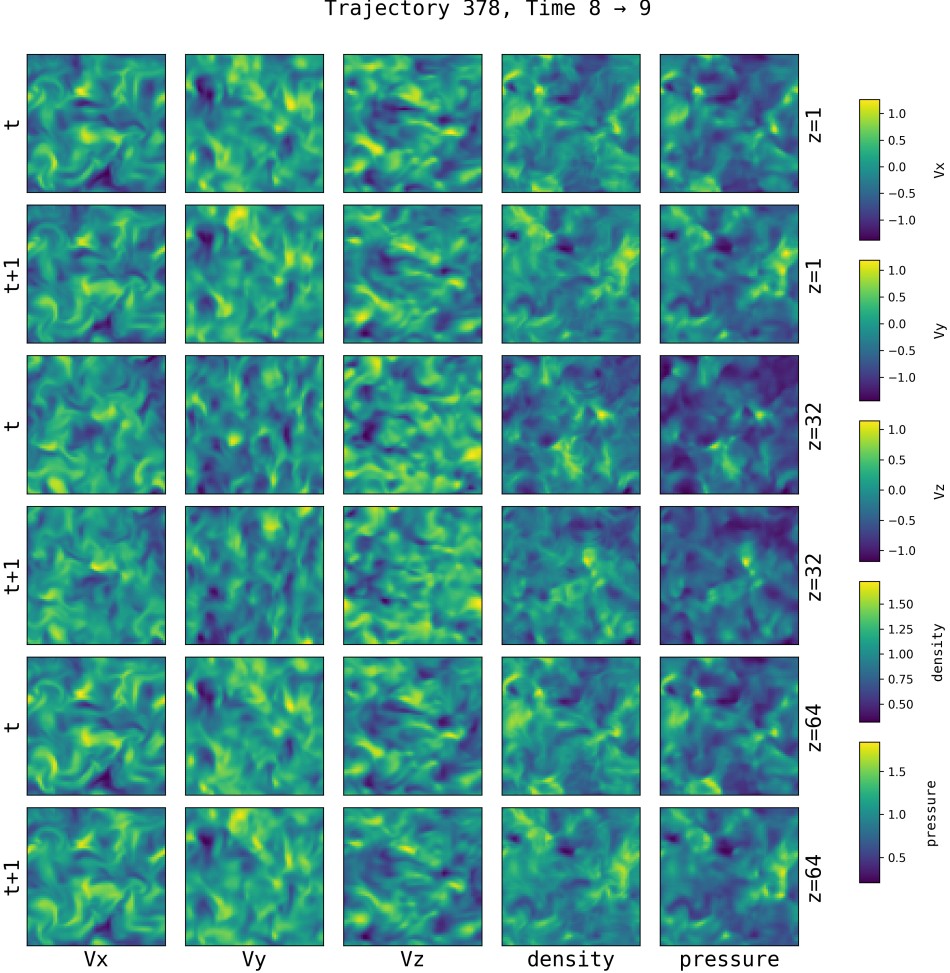

Figure 8: Example of two consecutive time steps from a three-dimensional Navier-Stokes simulation, visualized across different z-slices for all physical fields.

## C    TRAINING DETAILS

**Data Preparation.**    To minimize the one-step loss $\mathcal{L} = ||U_{t+1} - \mathcal{G}_\theta(U_t)||_2^2$, we require input-output pairs. Consistent with prior work (Li et al., 2021), we set the prediction step size $\Delta t_p = 0.8$ and use residual prediction ($\mathcal{G}_\theta(\mathbf{U}_n) \approx \mathbf{U}_{n+1} - \mathbf{U}_n$) to balance short-term (one-step loss) and long-term (trajectory) performance. We also spatially downsample to 256 points. We scale each target by dividing by the maximum value across all trajectories, time steps, and spatial points; we found this to perform marginally better than normalizing to unit standard deviation.

**Neural Simulator Architectures.**

**Fourier Layer.**    The Fourier layer transforms the input into the frequency domain using a fast Fourier transform (FFT), applies a truncated linear transformation to selected Fourier modes, and then maps the result back to the spatial domain via an inverse FFT. This spectral transformation is typically combined with a skip connection consisting of a point-wise convolution, a bias term, and an activation function. Formally, for an input $\mathbf{x} \in \mathbb{R}^n$, the layer computes:

$$\mathrm{F}(\mathbf{x}) = \sigma(\mathcal{F}^{-1}(R \cdot \mathcal{F}(\mathbf{x})) + W\mathbf{x} + \mathbf{b}),$$

where $\mathcal{F}$ and $\mathcal{F}^{-1}$ denote the FFT and inverse FFT respectively, $R : \mathbb{R}^n \to \mathbb{R}^{n'}$ is a learned linear transformation in frequency space, $W : \mathbb{R}^n \to \mathbb{R}^{n'}$ represents a point-wise convolution, and $\mathbf{b}$ is a bias term.

Several variations of the Fourier layer have been proposed. One such variant (Tran et al., 2021) modifies the layer by introducing a residual connection and a two-layer feedforward network, while omitting the point-wise convolution and bias term:

$$\mathrm{F}(\mathbf{x}) = \mathbf{x} + \sigma(W_2 \sigma(W_1 \mathcal{F}^{-1}(R \cdot \mathcal{F}(\mathbf{x})) + \mathbf{b}_1) + \mathbf{b}_2).$$

In our early experiments, this modification did not yield noticeable improvements. We also explored simply adding a skip connection without the feedforward block and inserting normalization layers at various points in the architecture, but these did not result in noticeable improvements.

**RecurrFNO**    The Fourier Neural Operator is made of a point-wise lifting layer, followed by a sequence of Fourier layers, and then a point-wise projection. The Recurrent Depth Simulator (RecurrSim) with Fourier layers can be interpreted in two ways: 1) RecurrFNO `wo/ EncDec`, a point-wise lifting layer encoder, followed by a sequence of Fourier layers (that make up the recurrent-depth block), and then a point-wise projection layer decoder, or 2) RecurrFNO `w/ EncDec`, where the first Fourier layer is part of the encoder and the last Fourier layer is part of the decoder. We find that RRecurrFNO `w/ EncDec` often leads to more consistent and superior performance.

**RecurrSim.**    The Recurrent Depth Simulator is a highly flexible framework. Each component—the encoder, recurrent-depth block, and decoder—may be instantiated with any layer(s) depending on the task. For example, in problems with periodic boundaries and a requirement of parameter efficiency, where the Fourier Neural Operator would typically shine, Fourier layers can be used. On the other hand, if the goal is to develop a foundation model for physics on irregular meshes, where one might use a graph-based encoder, with an attention-based bottleneck, and a graph-based decoder, the RecurrSim framework can be configured accordingly. With just a few additional lines of code, RecurrSim enables explicit control over the accuracy-cost trade-off (see Appendix J for pseudocode).

Fourier- and attention-based layers are well-suited for recurrent-depth blocks due to their ability to model infinite receptive fields. In contrast, convolutional-based layers have a fixed receptive field that grow with the depth. For example, a standard convolutional layer in `PyTorch` with `kernelsize=3`, `dialation=1`, and $stride = 1$ has a receptive field of size 3. Stacking two such layers increases he receptive field to 5—capturing the center point and two neighboring point on each side. More generally, the receptive field after stacking $L$ such layers is given by $L \cdot (\texttt{kernelsize}/2) + 1$. To achieve a receptive field of size 64, to effectively model the Burgers equation, one would need to stack 63 layers. In RecurrSim, where $K = 1$ could be sampled, 63 layers would need to be distributed across the encoder, recurrent-depth block, and decoder. To mitigate

this, some alternatives can be considered to expand the receptive field more efficiently: increasing the kernel size, incorporating attention-based layers, or adding downsampling blocks.

**FNO-DEQ.** Similarly to Marwah et al. (2023), we use Anderson acceleration with a maximum of 16 iterations. For the backward pass of the DEQ layer, we follow the phantom gradient approach proposed by Geng et al. (2021), using parameters $s = 3$ and $\tau = 0.8$. To match the parameter count of RecurrFNO, we employ a 1D FNO with 8 layers and 120 channels.

**ACDM.** We follow the original setup from Kohl et al. (2023), using a linear scheduler and training with a maximum of 50 diffusion steps. For conditioning, we concatenate the snapshot from the previous time step, i.e., the solution $u_t$ when predicting $u_{t+1}$. To ensure a fair comparison, we condition only on $u_t$ and do not include earlier time steps.

**PDE-Refiner** We use the same scheduler proposed by Lippe et al. (2023), with $\sigma_{\min}^2 = 2 \cdot 10^{-7}$ and $K = 10$. Following a similar approach to Kohl et al. (2023), we implement the following algorithm from scratch:

---

**Algorithm 3** PDE-Refiner: Training and Inference Procedures

---

1: **procedure** TRAINSTEP($u_t, u_{\text{prev}}$)
2:      $k \leftarrow$ random integer in $[0, \text{num\_steps}]$
3:      **if** $k = 0$ **then**
4:          pred $\leftarrow$ NeuralOperator(zeros\_like($u_t$), $u_{\text{prev}}$, $k$)
5:          target $\leftarrow u_t$
6:      **else**
7:          noise\_std $\leftarrow$ min\_noise\_std$^{k/\text{num\_steps}}$
8:          noise $\leftarrow$ randn\_like($u_t$)
9:          $u_{t,\text{noised}} \leftarrow u_t + \text{noise} \cdot \text{noise\_std}$
10:         pred $\leftarrow$ NeuralOperator($u_{t,\text{noised}}$, $u_{\text{prev}}$, $k$)
11:         target $\leftarrow$ noise
12:      **end if**
13:      loss $\leftarrow$ MSE(pred, target)
14:      **return** loss
15: **end procedure**

16: **procedure** PREDICTNEXTSOLUTION 1($u_{\text{prev}}$)
17:      $u_{\hat{t}} \leftarrow$ NeuralOperator(zeros\_like($u_{\text{prev}}$), $u_{\text{prev}}$, 0)
18:      **for** $k = 1$ **to** num\_steps **do**
19:          noise\_std $\leftarrow$ min\_noise\_std$^{k/\text{num\_steps}}$
20:          noise $\leftarrow$ randn\_like($u_t$)
21:          $u_{\hat{t},\text{noised}} \leftarrow u_{\hat{t}} + \text{noise} \cdot \text{noise\_std}$
22:          pred $\leftarrow$ NeuralOperator($u_{\hat{t},\text{noised}}$, $u_{\text{prev}}$, $k$)
23:          $u_{\hat{t}} \leftarrow u_{\hat{t},\text{noised}} - \text{pred} \cdot \text{noise\_std}$
24:      **end for**
25:      **return** $u_{\hat{t}}$
26: **end procedure**

---

Algorithm 3 is taken from Lippe et al. (2023), and the number of inference num\_steps is fixed at test time. To adapt the original algorithm, we investigated two variations: Algorithm 4 and 5. When $\bar{K} = \text{num\_steps}$, both methods recover the original procedure proposed in Lippe et al. (2023).

The first variation, Algorithm 4, adjusts the noise scheduler based on the number of inference steps. However, this strategy only performs well when the number of steps matches the training setup. To address this limitation, we introduce Algorithm 5, which retains the noise scheduler from training while allowing the number of inference steps to vary. This consistency in noise levels enhances stability and performance by preserving the distribution the network was trained on.

---

**Algorithm 4** Predict Next Solution 1

---

**procedure** PREDICTNEXTSOLUTION($u_{\text{prev}}$)
    $u_{\hat{t}} \leftarrow$ NeuralOperator(zeros_like($u_{\text{prev}}$), $u_{\text{prev}}$, 0)
    **for** $k = 1$ **to K do**
        noise_std $\leftarrow$ min_noise_std$^{k/\mathbf{K}}$
        noise $\leftarrow$ randn_like($u_t$)
        $u_{\hat{t},\text{noised}} \leftarrow u_{\hat{t}} +$ noise $\cdot$ noise_std
        pred $\leftarrow$ NeuralOperator($u_{\hat{t},\text{noised}}$, $u_{\text{prev}}$, $k$)
        $u_{\hat{t}} \leftarrow u_{\hat{t},\text{noised}} -$ pred $\cdot$ noise_std
    **end for**
    **return** $u_{\hat{t}}$
**end procedure**

---

**Algorithm 5** Predict Next Solution 2

---

1: **procedure** PREDICTNEXTSOLUTION($u_{\text{prev}}$)
2:     $u_{\hat{t}} \leftarrow$ NeuralOperator(zeros_like($u_{\text{prev}}$), $u_{\text{prev}}$, 0)
3:     **for** $k = 1$ **to K do**
4:         noise_std $\leftarrow$ min_noise_std$^{k/\text{num\_steps}}$
5:         noise $\leftarrow$ randn_like($u_t$)
6:         $u_{\hat{t},\text{noised}} \leftarrow u_{\hat{t}} +$ noise $\cdot$ noise_std
7:         pred $\leftarrow$ NeuralOperator($u_{\hat{t},\text{noised}}$, $u_{\text{prev}}$, $k$)
8:         $u_{\hat{t}} \leftarrow u_{\hat{t},\text{noised}} -$ pred $\cdot$ noise_std
9:     **end for**
10:     **return** $u_{\hat{t}}$
11: **end procedure**

---

**Optimization.** All optimization hyperparameters are listed in Table 6 and remain fixed across all experiments, except where explicitly stated. We train each model for 100 epochs using the AdamW optimizer (Loshchilov & Hutter, 2019), starting with a learning rate of $3 \times 10^{-4}$ and a weight decay of $1 \times 10^{-5}$. A cosine annealing schedule is applied to gradually reduce the learning rate to $3 \times 10^{-6}$ (Loshchilov & Hutter, 2017). In early experiments, we observed that using a higher initial learning rate (e.g., $1 \times 10^{-3}$) led to less consistent performance, though it occasionally improved performance (Sohl-Dickstein, 2024).

| Hyperparameter | Value |
|---|---|
| Epochs | 100 |
| Batch Size | 256 [2] |
| Optimizer | AdamW |
| Starting Learning Rate | $3 \times 10^{-4}$ |
| Weight Decay | $1 \times 10^{-5}$ |
| Scheduler | Cosine Annealing |
| Ending Learning Rate | $3 \times 10^{-6}$ |

Table 6: Optimization hyperparameters used in all experiments.

---

[2]For the three-dimensional experiments, we use a batch size of 32, and perform gradient accumulation to have an effective batch size of 256.

## D  BACKPROPAGATION WINDOW

During training, the recurrent-depth block is repeated $K$ times in the forward pass, after which gradients are propagated backward through the same computation. If $K$ is large, which could happen because $K$ is drawn from a long-tailed distribution, the backward pass must retain every intermediate activation, quickly exhausting GPU memory. To cap the memory usage, we use truncated backpropagation-through-*depth* with a fixed backpropagation window $B$: gradients are backpropagated through at most the last $B$ steps, and earlier steps are treated as constants. This bounds memory at $O(B)$ independent of $K$. In this experiment, we study it truncated backpropagation-through-*depth* is viable and the effect of different backpropagation windows.

**Experimental Setup.** We conduct experiments on three datasets: Burgers, long-horizon KdV, and long-horizon KS. We train a recurrent depth simulator with a point-wise lifting layer, a recurrent-depth block with a single Fourier layer, and a point-wise projection layer with $\sim$ 1M parameters. We set $\bar{K} = 32$, and the backpropagation window is swept over $B \in \{1, 2, 4, 16, 32\}$. With $B = 1$ the compute for the forward pass is equivalent to a Fourier layer with 33 layers, but the backward pass stores only a single activation; with $B = 32$ the backward pass stores every activation whenever $K \leq 32$ and the last 32 when $K > 32$. This would be infeasible for higher-dimensional problems.

**Results.** We report the trajectory errors in Table 7. Across all equation $B = 1$ performs worst and moving from $B = 1$ to $B = 2$ yields the largest gain, and improvements largely saturate by $B = 4$. Beyond $B = 4$, larger windows offer only marginal benefit while reinstating a substantial memory cost. Note that although trajectory error is not the preferred metric for KS, the same saturation is evident. Based on these results, and to balance accuracy and memory, we set $B = 4$ in all main experiments.

| Backpropagation Window $B$ | Burgers | Korteweg-De Vries | Kuramoto-Sivashinsky |
|---:|---|---|---|
| 1 | 0.0849 | 0.1046 | 1.6341 |
| 2 | 0.0315 | 0.0522 | 1.4097 |
| 4 | 0.0199 | 0.0317 | 1.3972 |
| 16 | 0.0181 | 0.0302 | 1.3960 |
| 32 | 0.0178 | 0.0298 | 1.3910 |

Table 7: Impact of the back-propagation window $B$ on trajectory error. Accuracy improves sharply up to $B = 4$ and then plateaus.

## E  DISTRIBUTION PARAMETER $\bar{\text{K}}$

The distribution parameter $\bar{K}$ controls the expected number of recurrent steps during training. Setting $\bar{K}$ too low shortens training time but may leave the model under-exposed to large $K$ values during inference; setting it too high increases training time. In this experiment, we wish to identify the optimal $\bar{K}$.

**Experimental Setup.** We conduct experiments on three datasets: Burgers, long-horizon KdV, and long-horizon KS. We train a recurrent depth simulator with a point-wise lifting layer, a recurrent-depth block with a single Fourier layer, and a point-wise projection layer with $\sim$ 1M parameters. The backpropagation window is fixed at $B = 4$, and $\bar{K}$ is swept over $\{1, 2, 4, 8, 16, 32, 64, 128\}$. Doubling $\bar{K}$ roughly doubles the forward cost, yet backward memory remains capped by $B$; for instances, $\bar{K} = 8$ matches the forward FLOPS of an 8-layer FNO but the truncated-backpropagation-through-*depth* keeps the backward pass FLOPs as cheap as a 4-layer FNO. After training, each model is evaluated at all values $K \in [1, 2\bar{K}]$ and we report the lowest trajectory error achieved.

**Results.** Figure 9-11 plot trajectory error as a function of $\bar{K}$. Increasing $\bar{K}$ consistently lowers the best achievable trajectory error, but we observe diminishing returns beyond $\bar{K} \approx 32$. We also notice that models trained with larger $\bar{K}$ underperform with small $K$ values (see Figure 20). In other words, the additional training compute shifts the accuracy-cost curve to the right and gains appear only once $K$ is allowed to grow. Based on these results, we set $\bar{K} = 32$ in our main experiments as it captures the bulk of the benefit of high-compute settings while leaving the model competitive in low-compute settings.

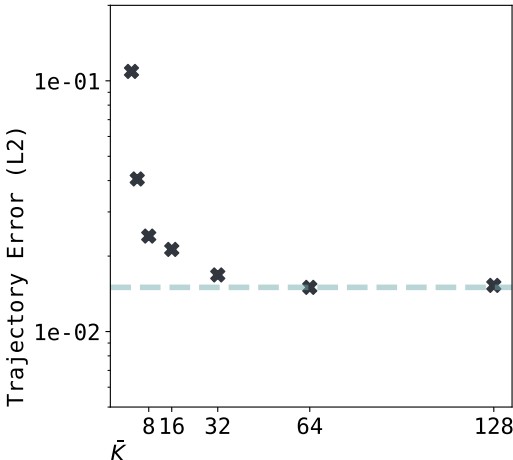

Figure 9: Choosing the distribution parameter $\bar{K}$ on the Burgers dataset.

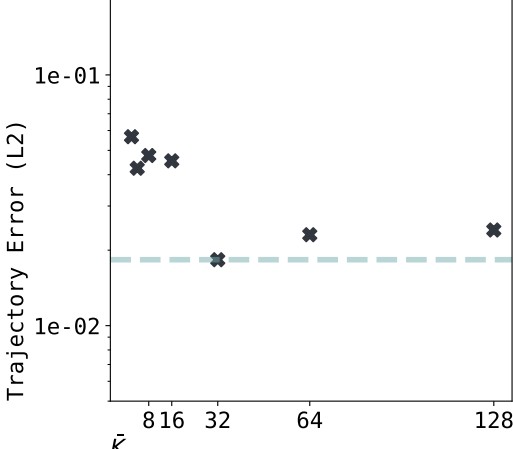

Figure 10: Choosing the distribution parameter $\bar{K}$ on the KdV dataset.

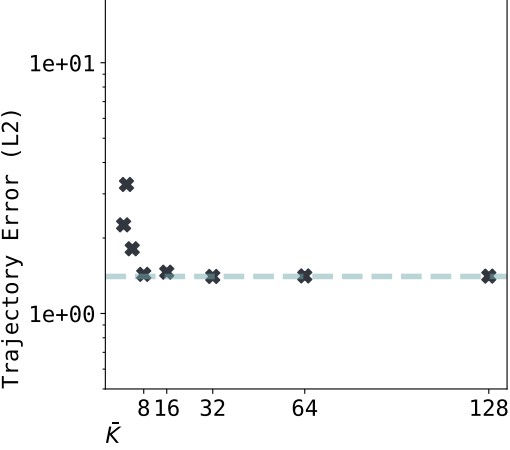

Figure 11: Choosing the distribution parameter $\bar{K}$ on the KS dataset.

## F  FIXED POINT

At test-time, the user is free to select the number of recurrent iterations $K$ according to their desired accuracy and available computational resources. Figure 2 (Top) and Figure 3 demonstrate that increasing recurrent iterations $K$ yields more accurate trajectories. To understand this behavior, we investigate how the latent vectors $z_k$ evolve with increasing recurrent iterations $K$ and whether they converge to fixed points $z^*$.

**Experimental Setup.**  We conduct experiments on two datasets: Burgers and long-horizon KdV. the trained RecurrFNO w/ `EncDec` models from Section 4.2 with $K = 32$. For each timestep in the trajectory, we measure the Euclidean distance between latent vector $z_k$ and approximate fixed point $z^* = z_K$ for all $k \in [0, K]$. We consider the convergence criterion $||z_k - z^*|| < \epsilon$ for a chosen tolerance $\epsilon$.

**Results.**  Figure 12 shows predicted trajectories for Burgers (Top Left) and long-horizon KdV (Top Right). The heatmaps (Bottom) show the Euclidean distance between latent vector $z_k$ and approximate fixed point $z^* = z_K$ for all $k \in [0, K]$ across time. For both equations, we see successive recurrent block applications generate increasingly similar latent vectors $||z_{k+1} - z^*|| < ||z_k - z^*||$, eventually satisfying our convergence criterion $||z_k - z^*|| < 1e^{-2}$. This indicates practical convergence to a fixed point. Remarkably, the convergence behavior reflects the underlying physics. For the Burgers equation, which exhibits two distinct regimes, an early nonlinear phase that develops shocks and a later viscous-dominated phase, we find that more recurrent iterations are required for convergence during the shock-formation period. This behavior matches that of adaptive numerical methods. For long-horizon KdV, we find spikes throughout the trajectory due to the more difficult task of soliton interactions. These results demonstrate that RecurrSim's latent vector does converge toward a fixed point and that the model behavior is aligned with the underlying physics.

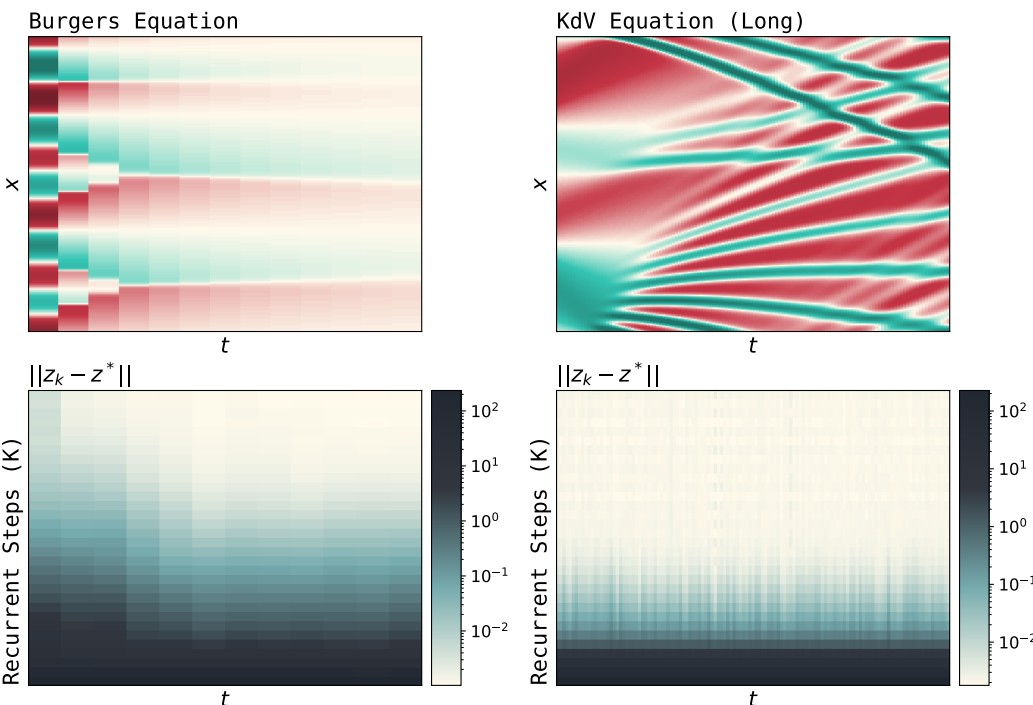

Figure 12: Convergence analysis showing predicted trajectories (top) and heatmaps of $||z_k - z^*||$ (bottom) for Burgers (left) and KdV (right) equations. Darker colors indicate larger distances from the approximate fixed point.

# G  INITIAL LATENT DISTRIBUTION

RecurrSim's initial latent vector $\mathbf{z}_0$ is drawn from a fixed distribution $p(\mathbf{z})$ and iteratively updated through the recurrent-depth block. A good initial latent vector could potentially require fewer iterative updates to reach better latent vectors for decoding. In this experiment, we study how different initial latent distributions affect performance across varying recurrent iterations $K$.

**Experimental Setup.**   We conduct experiments on two datasets: Burgers and long-horizon KdV. We train a recurrent depth simulator with a point-wise lifting layer, a recurrent-depth block with a single Fourier layer, and a point-wise projection layer with $\sim$ 1M parameters. We set the backpropagation window $B = 4$ and $\bar{K} = 32$. We ablate four different initial latent vector choices:

1. Standard normal: $\mathbf{z}_0 \sim \mathcal{N}(\mathbf{0}, \mathbf{I})$
2. Zero initialization: $\mathbf{z}_0 = \mathbf{0}$ (Dirac delta at zero, $\delta_{\mathbf{0}}$)
3. Conditioning initialization: $\mathbf{z}_0 = \mathbf{c}$ (Dirac delta at conditioning vector, $\delta_{\mathbf{c}}$)
4. Learnable initialization: $\mathbf{z}_0 = \mathbf{z}_\theta$ (learned parameter independent of input)

After training, we evaluate the model at recurrent iterations $K \in \{2, 4, 8, 16, 32\}$ and report the trajectory error.

**Results.**   Table 8and Table 9 show trajectory errors for all four initial latent vectors across different values of $K$. For Burgers equation, standard normal and zero initialization ($\delta_{\mathbf{0}}$) perform the best. At $K = 2$, the learnable latent vector performs poorly, but by $K = 16$, all initial latent vectors reach similar performance. For long-horizon KdV, zero initialization performs best, but we observe similar behavior: all initial latent vectors reach similar performance. Across both datasets and all recurrent iterations $K \in \{2, 4, 8, 16, 32\}$, we find zero initialization provides the most consistent performance. However, standard normal performs comparably and is consistent with DEQ and diffusion models. These results demonstrate while the initial latent distribution can play a part early on, given sufficient recurrent iterations, different distributions reach similar performance.

| Recurrent Iteration $K$ | $\mathbf{z}_0 \sim \mathcal{N}(\mathbf{0}, \mathbf{I})$ | $\mathbf{z}_0 \sim \delta_{\mathbf{0}}$ | $\mathbf{z}_0 \sim \delta_{\mathbf{c}}$ | $\mathbf{z}_0 = \mathbf{z}_\theta$ |
|---|---|---|---|---|
| 2 | *0.3609* | **0.2183** | 0.5553 | 1.5921 |
| 4 | *0.0968* | **0.0789** | 0.1287 | 0.1500 |
| 8 | **0.0192** | *0.0199* | 0.0293 | 0.0205 |
| 16 | *0.0101* | **0.0101** | 0.0132 | 0.0102 |
| 32 | *0.0100* | **0.0099** | 0.0129 | 0.0102 |

Table 8: Trajectory error on Burgers for four initial latent distributions across five recurrent iterations. Best result in each row is **bold**, second-best *italic*.

| Recurrent Iteration $K$ | $\mathbf{z}_0 \sim \mathcal{N}(\mathbf{0}, \mathbf{I})$ | $\mathbf{z}_0 \sim \delta_{\mathbf{0}}$ | $\mathbf{z}_0 \sim \delta_{\mathbf{c}}$ | $\mathbf{z}_0 = \mathbf{z}_\theta$ |
|---|---|---|---|---|
| 2 | **0.5892** | 1.2367 | 1.8251 | *0.7051* |
| 4 | 0.1570 | **0.1176** | 0.2412 | *0.1181* |
| 8 | *0.0188* | **0.0167** | 0.0194 | 0.0201 |
| 16 | 0.0187 | **0.0164** | *0.0176* | 0.0205 |
| 32 | 0.0187 | **0.0164** | *0.0177* | 0.0205 |

Table 9: Trajectory error on long-horizon KdV for four initial latent distributions across five recurrent iterations. Best result in each row is **bold**, second-best *italic*.

# H    MERGING

At each recurrent step, the recurrent depth simulator must merge the condition vector $\mathbf{c}$ with the latent vector $\mathbf{z}_k$. We consider six merging methods of increasing capacity. **Add** simply sums the two vectors. **Add$_\mathbf{s}$** introduces two learnable parameters $\alpha$ and $\beta$ ($\mathbf{z}'_k = \alpha \mathbf{c} + \beta \mathbf{z}_k$). **Add$_\mathbf{e}$** generalizes this to element-wise vectors $\boldsymbol{\alpha}$ and $\boldsymbol{\beta}$ ($2 \times$ `hiddenchannels` additional trainable parameters). **Projection** concatenates $[\mathbf{c}, \mathbf{z}_k]$ and applies a point-wise linear map ($2 \times$ `hiddenchannels` $\times$ `hiddenchannels` additional trainable parameters); **Projection$_\mathbf{I}$** uses the same layer but is initialized with 1s along the diagonals and 0s everywhere else, so that it is equivalent to **Add$_\mathbf{e}$** at initialization but with increased capacity. **Concat** feeds the raw concatenation into the first layer (in the recurrent-depth block), doubling its input channels, and thus, trainable parameters. In this experiment, our goal is to test these merging methods.

**Experimental Setup.**    All experiments run on the one-dimensional Burgers equation. The base architecture is fixed—a point-wise lift, a single Fourier layer encoder, a one-layer Fourier recurrent block, and a Fourier decoder with point-wise projection—trained with $\bar{K} = 32$ and backpropagation window $B = 4$. We sweep five parameter budgets $\{0.2\text{M}, 0.5\text{M}, 1.0\text{M}, 2.0\text{M}, 4.0\text{M}\}$ by scaling channel width, and implement each of the six merging methods at every budget. After training, each model is evaluated at all values $K \in [1, 2\bar{K}]$ and we report the lowest trajectory error achieved.

**Results.**    Table 10 reports the lowest trajectory error for every configuration. The three addition variants perform almost identically and improve monotonically with parameter count. The **Projection** variant lags behind, but when initialized with 1s along the diagonals (**Projection$_\mathbf{I}$**), it matches or exceeds the additional family. **Concat** attains the lowest error overall, but at the price of $\sim 33\%$ extra parameters in the recurrent-block's first layer; we hypothesize that part of its gain stems from increased model size rather than a superior merging mechanism.

| **Parameters** | **Add** | **Add$_\mathbf{s}$** | **Add$_\mathbf{e}$** | **Projection** | **Projection$_\mathbf{I}$** | **Concat** |
|---|---|---|---|---|---|---|
| $\sim 0.2$M | 0.0234 | 0.0230 | *0.0229* | 0.0240 | 0.0240 | **0.0214** |
| $\sim 0.5$M | 0.0176 | 0.0173 | 0.0172 | 0.0223 | **0.0135** | *0.0146* |
| $\sim 1.0$M | 0.0129 | *0.0126* | *0.0126* | 0.0169 | **0.0101** | 0.0151 |
| $\sim 2.0$M | 0.0116 | 0.0115 | 0.0115 | 0.0094 | *0.0093* | **0.0090** |
| $\sim 4.0$M | 0.0100 | *0.0098* | 0.0099 | 0.0110 | 0.0100 | **0.0083** |

Table 10: Trajectory error on Burgers for six merging methods across five parameter budgets. Best result in each row is **bold**, second-best *italic*.

# I MORE EXPERIMENTS

## I.1 EXPERIMENT: ACCURACY-COST TRADE-OFF (EXTENDED)

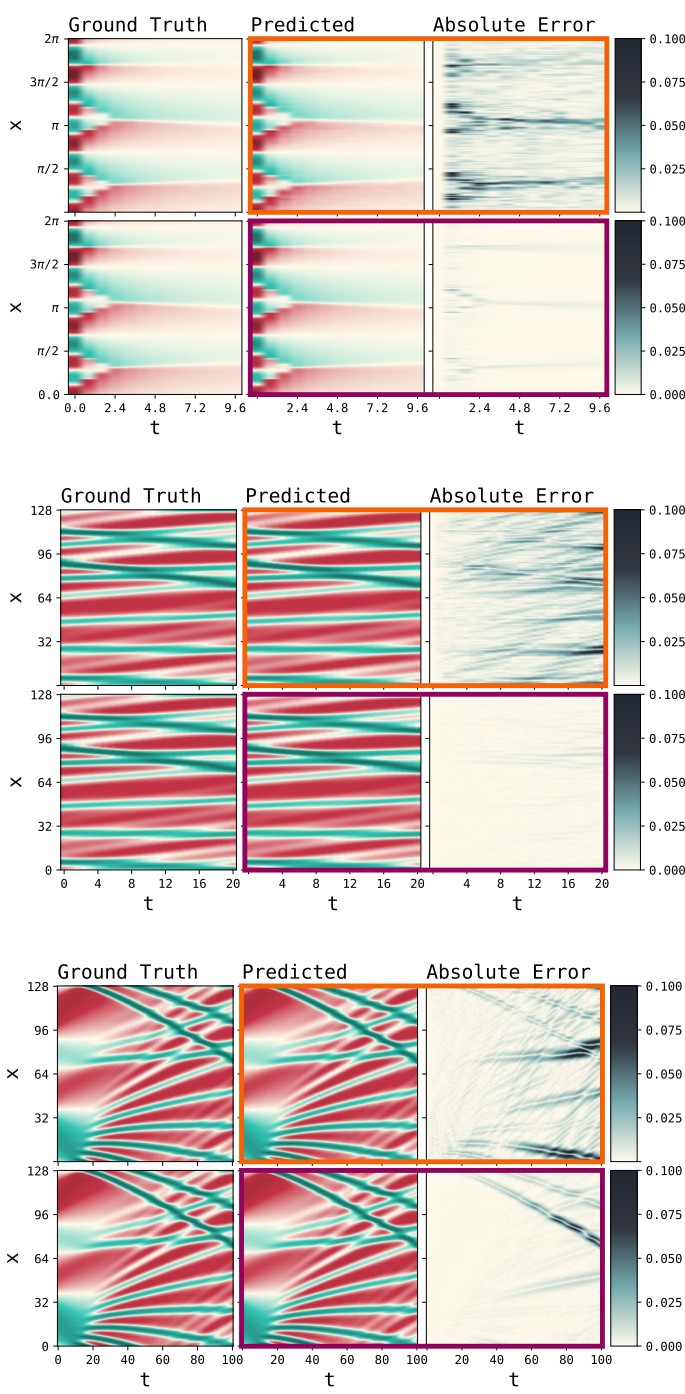

Figure 13: Supplementary visualization for Figure 2 (Bottom) showing ground truth, predictions, and absolute errors with colorbars. **Top:** Burgers, **Middle:** short-horizon KdV, **Bottom:** long-horizon KdV. Within each equation: low-compute $K = 4$ (orange border) and high-compute $K = 16$ (purple border). RecurrSim maintains physical fidelity at both computational budgets, with errors decreasing as $K$ increases.

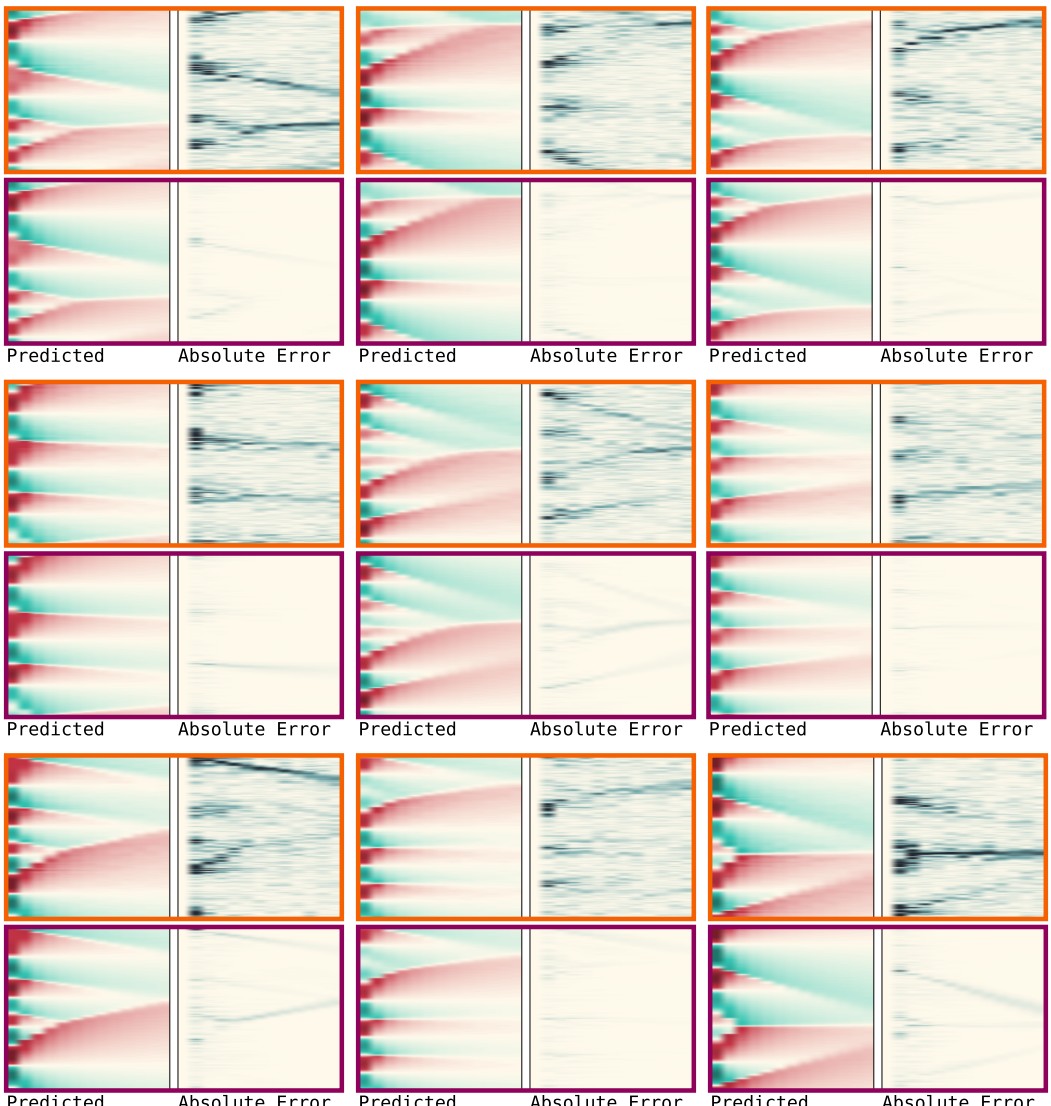

Figure 14: Burgers: Trajectories at $K = 4$ (orange) and $K = 16$ (purple).

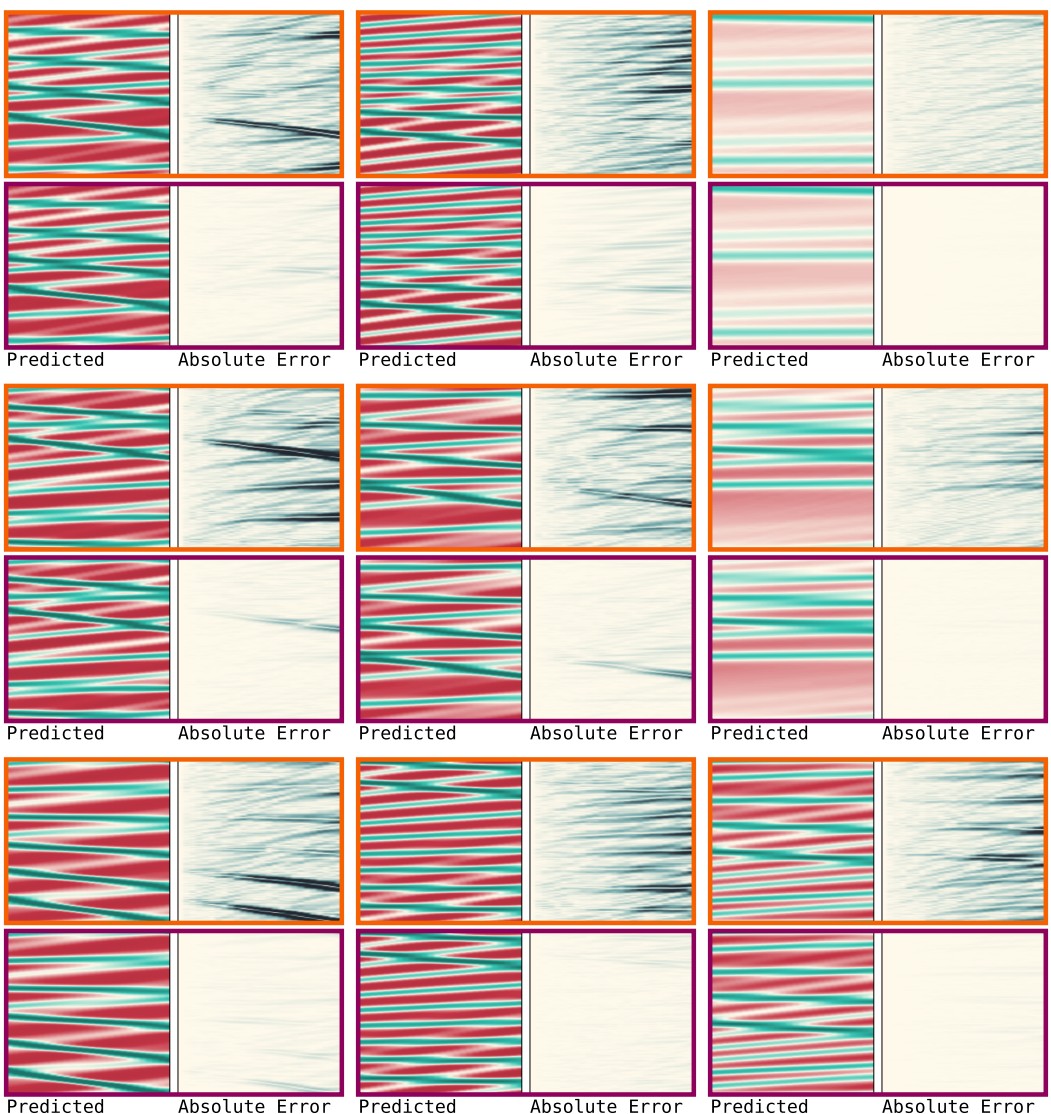

Figure 15: Short-Horizon KdV: Trajectories at $K = 4$ (orange) and $K = 16$ (purple).

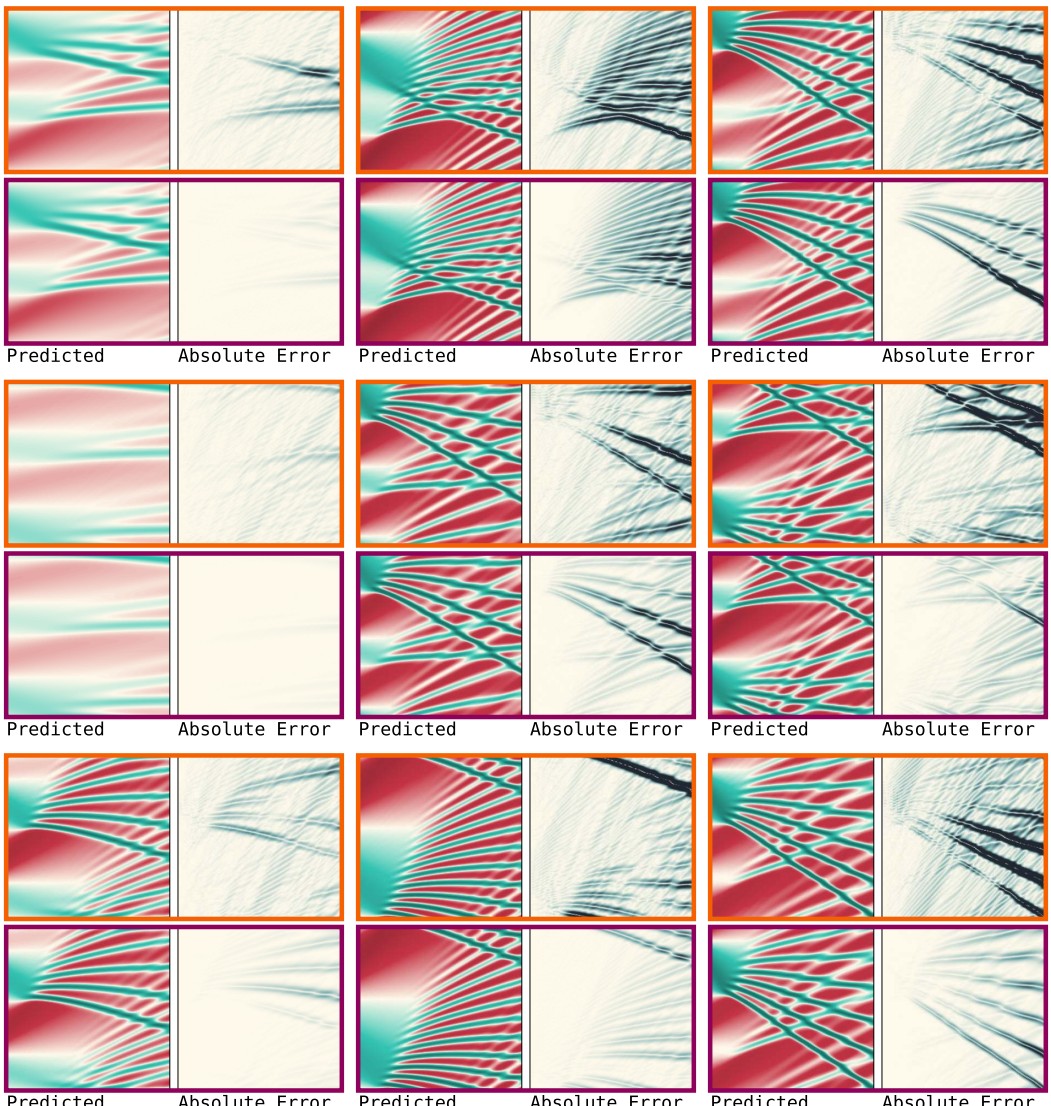

Figure 16: Long-Horizon KdV: Trajectories at $K = 4$ (orange) and $K = 16$ (purple).

## I.2 EXPERIMENT: ALTERNATIVES (EXTENDED)

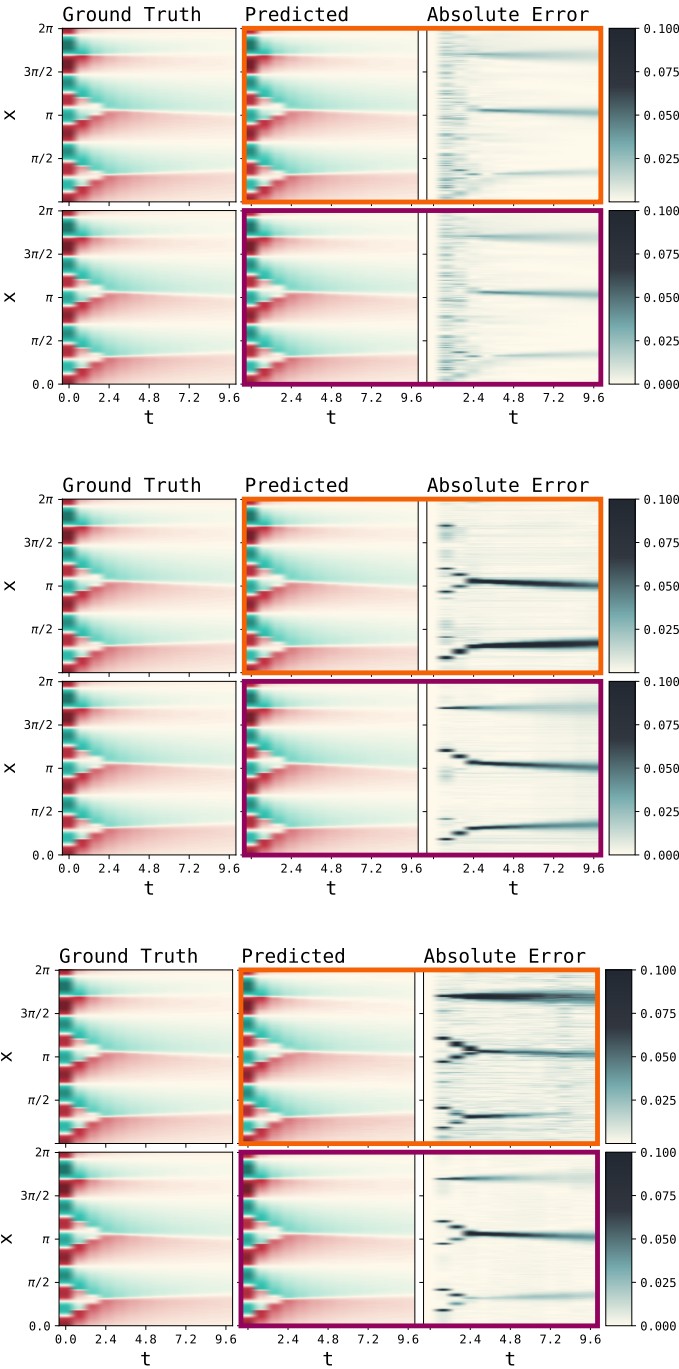

Figure 17: Performance of baseline methods on Burgers equation. **Top:** FNO-DEQ, **Middle:** ACDM, **Bottom:** PDE-Refiner. Within each method: low-compute setting (orange border) and high-compute setting (purple border). FNO-DEQ shows minimal improvement with additional compute. ACDM and PDE-Refiner fail to accurately reproduce shock structures at low compute and show limited improvement at high compute.

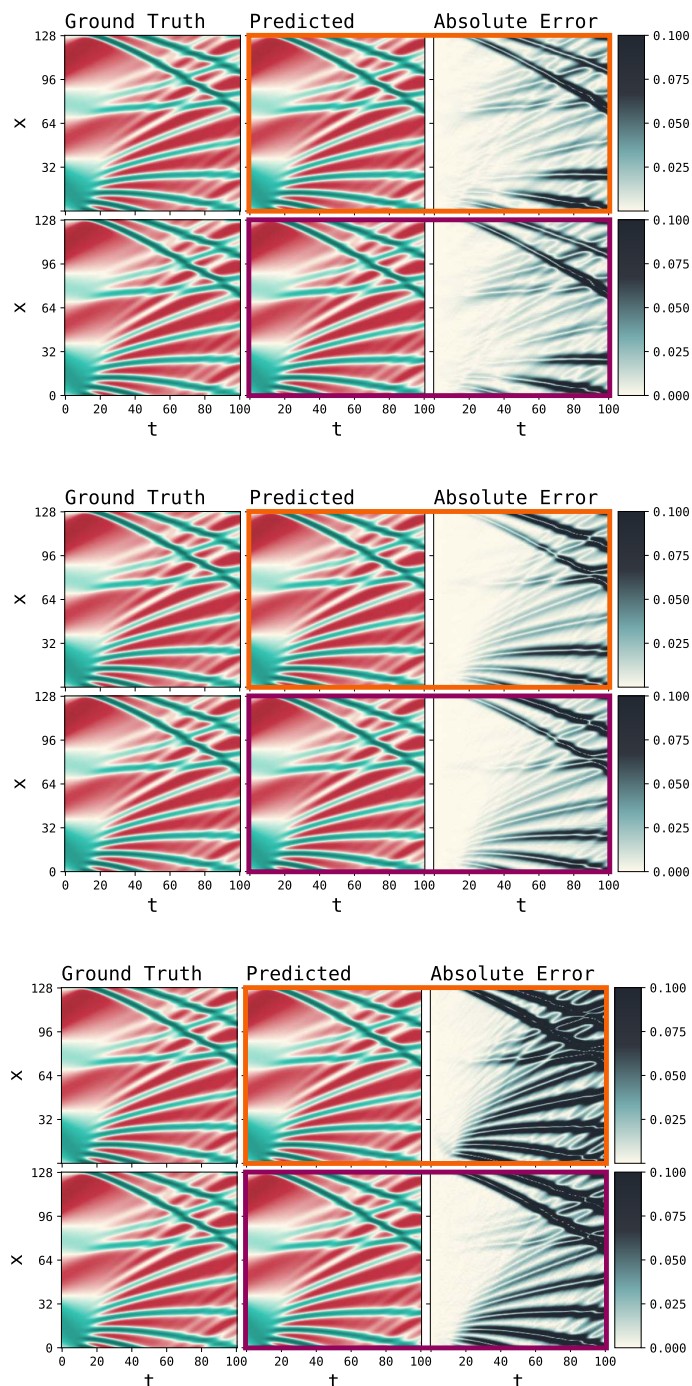

Figure 18: Performance of baseline methods on long-horizon KdV equation. **Top:** FNO-DEQ, **Middle:** ACDM, **Bottom:** PDE-Refiner. Within each method: low-compute setting (orange border) and high-compute setting (purple border). All three baselines fail to accurately reproduce solitons.

Table 11 summarizes the best performance achieved by each method across all recurrent iterations $K$ (mean $\pm$ standard deviation over 3 seeds). RecurrFNO consistently outperforms all baselines across both trajectory error (Burgers, KdV) and correlation horizons (KS), achieving 82% lower error than the best baseline on Burgers, 21% lower on KdV, and up to 98% longer correlation horizons on KS.

| Model | Burgers MSE($\downarrow$) | KdV MSE($\downarrow$) | KS Avg. Corr. $\alpha = 0.8(\uparrow)$ | KS Avg. Corr. $\alpha = 0.9(\uparrow)$ |
|---|---|---|---|---|
| RecurrFNO | **0.008 ± 0.001** | **0.022 ± 0.005** | **74.7 ± 4.2** | **69.2 ± 3.2** |
| FNO-DEQ | 0.045 ± 0.007 | 0.094 ± 0.028 | 41.9 ± 5.6 | 37.0 ± 4.6 |
| ACDM | 0.044 ± 0.004 | 0.043 ± 0.015 | 40.5 ± 9.3 | 35.6 ± 9.2 |
| PDE-Refiner | 0.161 ± 0.024 | 0.028 ± 0.010 | 61.6 ± 7.7 | 56.6 ± 8.8 |

Table 11: Best performance across all recurrent iterations $K$ for RecurrFNO and baseline adaptive-compute methods (mean $\pm$ std over 3 seeds).

On the chaotic Kuramoto-Sivashinsky dataset we replace trajectory error with the average and worst-case correlation horizon metrics. Figure 19 shows the behavior of the four adaptive-compute simulators across 30 correlation thresholds ($\alpha = 0.70 - 0.99$) and all inference depths $K \in \{1, \ldots, 16\}$. RecurrFNO (first column) shows the desired monotone pattern: both the average and the worst-cast correlation horizons rise steadily with $K$. FNO-DEQ delivers flat surfaces—its iterations leave the horizon essentially unchanged—so it cannot exploit extra compute. ACDM begins with short horizons, improves up to $K \approx 4$, and then flattens; only a narrow band of $K$ values is usable, limiting its test-time flexibility. PDE-Refiner gains up to $K \approx 8$ but then oscillates, making it hard to pick a reliable stopping point. Across both average and worst-case statistics RecurrSim attains the longest horizons and is the only model whose accuracy scales predictably with additional compute, confirming its advantage for controllable accuracy-cost trade-offs in chaotic regimes.

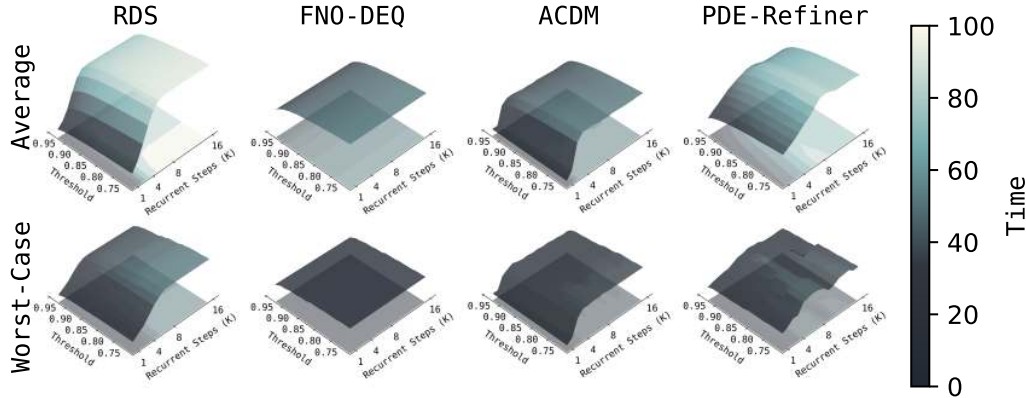

Figure 19: Kuramoto–Sivashinsky: average (top) and worst-case (bottom) correlation horizons and threshold $\alpha$ versus inference depth $K$.

### I.3 EXPERIMENT: HIGH-DIMENSIONAL SIMULATIONS AND TRANSFORMER VARIANTS (EXTENDED)

As shown in fig. 20, the two models present distinct trade-offs. When the number of recurrent steps during inference exceeds 8, the model trained with a higher FLOPs budget and a higher $\bar{K}$ yields significantly lower MSE. In contrast, for fewer than 8 recurrent steps, the model trained with a lower computational budget performs better.

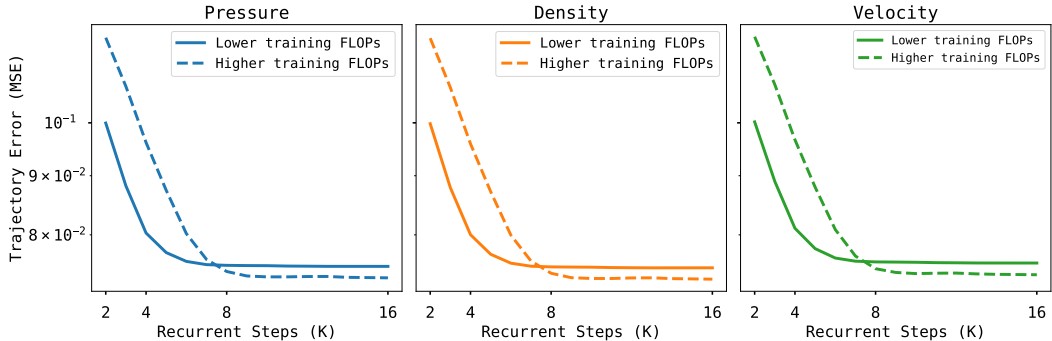

Figure 20: Trajectory error (MSE) over the number of recurrent steps $K$ for two RecurrFNO models, trained with lower and higher FLOPs budgets

Figure 21 shows the trajectory error as a function of recurrent steps $K$ for RecurrViT on the Active Matter dataset, evaluated at three different rollout horizons (steps 0:3, 0:6, and 0:12). The dashed lines represent the performance of a standard ViT baseline with comparable parameter count.

RecurrViT demonstrates monotonic improvement with increasing $K$ across all rollout horizons. For short rollout horizons (0:3), RecurrViT achieves 83% lower error than ViT (0.005 vs 0.029). For medium rollout horizons (0:6), RecurrViT achieves 90% lower error than ViT (0.013 vs 0.124), and already surpasses ViT performance by $K = 8$. For long rollout horizons (0:12), RecurrViT achieves 87% lower error than ViT (0.057 vs 0.432).

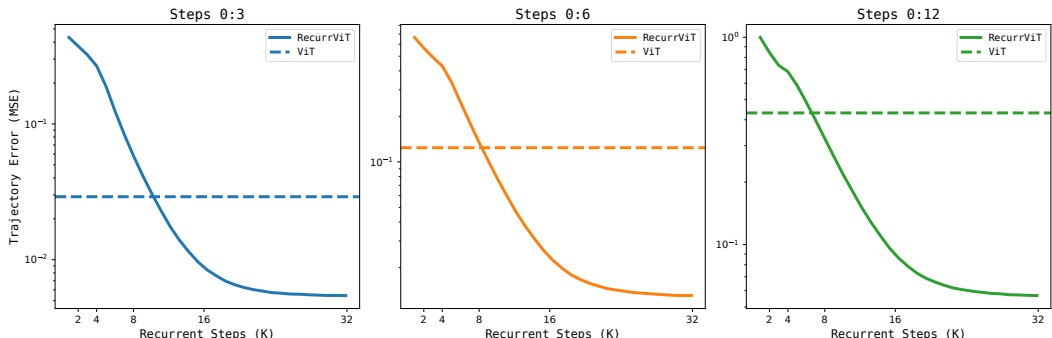

Figure 21: Trajectory error (MSE) over the number of recurrent steps $K$ for RecurrViT (solid lines) and standard ViT (dashed lines) on the Active Matter dataset, evaluated at three rollout horizons: steps 0:3 (left), 0:6 (middle), and 0:12 (right).

## J  PSEUDOCODE

```python
class Network(Module):
    def __init__(self):
        super().__init__()
        # Encoder Layer
        self.encoder = Layer()

        # Collect L Intermediate Layers
        layers = []
        for _ in range(L):
            layers.append(Layer())

        # Decoder Layer
        self.decoder = Layer()

    def forward(self, x):
        # Apply Encoder
        z = self.encoder(x)

        #####################
        ##### Main Block #####

        # Apply L Intermediate Layers
        for layer in self.layers:
            z = layer(z)

        ##### Main Block #####
        #####################

        # Apply Decoder
        x = self.decoder(z)
        return x
```

Listing 1: Pseudocode of a *standard* neural simulator. The neural simulator contains an encoder or lifting layer (`self.encoder`), L intermediate layers of any type (residual layers, Fourier layers, etc.), and an decoder or projection layer (`self.decoder`).

```python
class Network(Module):
    def __init__(self):
        super().__init__()
        # Encoder Layer
        self.encoder = Layer()

        # Collect L Intermediate Layers
        layers = []
        for _ in range(L):
            layers.append(Layer())

        # Decoder Layer
        self.decoder = Layer()

    def forward(self, x, K=None):
        # Apply Encoder
        c = self.encoder(x)

        #######################
        ##### Main Block #####

        # Sample Noise \w 'shape=x.shape'
        z = sample_noise()

        # During Inference:
        if not self.training:
            # Loop K Times
            for _ in range(K):
                # Concatenate x and z
                z = cat([c, z], dim=1)
                # Apply L Intermediate Layers
                for layer in self.layers:
                    z = layer(z)

        # During Training:
        if self.training:
            # Do Not Use Grad
            with no_grad():
                # Sample K (Using K_bar)
                K = sample_K()
                # Loop K - B Times
                for _ in range(K - B):
                    z = cat([c, z], dim=1)
                    for layer in self.layers:
                        z = layer(z)
            # Loop Remaining B Times
            for _ in range(B):
                z = cat([c, z], dim=1)
                for layer in self.layers:
                    z = layer(z)

        ##### Main Block #####
        #######################

        # Apply Decoder
        x = self.decoder(z)
        return x
```

Listing 2: Pseudocode of the Recurrent Depth Simulator—fewer than 20 new lines compared to a *standard* neural simulator. During inference, we apply the intermediate layers K times. During training, we apply the intermediate layers K - B times without gradient, and B times with gradient. Nothing else needs to change.

## K    EXTENDED RELATED WORK

**Deep Equilibrium Models.**    Deep Equilibrium Models (DEQs), introduced by Bai et al. (2019), are implicit, infinite-depth, weight-tied neural networks. A DEQ directly solves for the fixed point of a nonlinear transformation using any black-box root-finding algorithm and instead of backpropagating through each layer, which can be infeasible due to memory and numerical stability, the DEQ makes use of the Implicit Function Theorem to compute the gradients at the equilibrium—this approach has a constant memory requirement regardless of depth. Although the existence of the fixed point, or convergence to the fixed point, is not guaranteed; on large-scale language modeling tasks, Bai et al. (2019) demonstrated that DEQs can achieve performance comparable with state-of-the-art while using significantly less memory. Later, Bai et al. (2020) extended DEQs to large-scale computer vision tasks, showing similar performance and memory benefits. Subsequent research explored DEQs for various applications. Pokle et al. (2022) represent the entire sampling process in denoising diffusion implicit models as a single fixed-point system. Geng et al. (2023) distill diffusion models, directly from initial noise to the final image, into a DEQ. In inverse problems, Gilton et al. (2021) model a, potentially infinite, iterative reconstruction scheme as a DEQ. For partial differential equations, Pokle et al. (2022) propose FNO-DEQ, a DEQ variant with Fourier layers, to solve steady-state PDEs, showing improvements in accuracy and robustness to noise over baselines with four times as many parameters.

**Denoising Diffusion Models.**    First introduced by Sohl-Dickstein et al. (2015), diffusion models are probabilistic models with an iterative forward diffusion process and a learned reverse diffusion process. The forward process gradually adds noise to data until only noise remains, and the reverse process gradually removes noise to restore the original data. New samples are generated by sampling a noise vector and passing it through the reverse process. Ho et al. (2020) presented high-quality image synthesis results using diffusion models. Dhariwal & Nichol (2021) and Karras et al. (2022) made further progress leading to state-of-the-art results and widespread adoption. Diffusion models have been applied to image generation (Nichol et al., 2021; Ramesh et al., 2022; Saharia et al., 2022b), image inpainting and outpainting (Saharia et al., 2022a), super-resolution (Saharia et al., 2022c), audio generation (Chen et al., 2020; Kong et al., 2020), text generation (Austin et al., 2021), including large language (diffusion) models (Nie et al., 2025). In scientific domains, diffusion models have been applied to medium-range weather forecasting (Price et al., 2023), structure-based drug design (Schneuing et al., 2024), and stable materials generation (Yang et al., 2023). Kohl et al. (2023) demonstrated that diffusion models are viable for turbulent flow simulation. Their results show that diffusion models outperform, in terms of long-term accuracy and stability, more efficient (and more commonly used) neural simulators. Kohl et al. (2023) also compared against PDE-Refiner (Lippe et al., 2023), a diffusion-based multi-step refinement process, but found that PDE-Refiner is highly sensitive to hyperparameters, and in some cases, generated substantially worse results compared to other methods.

## L    EXTENDED DISCUSSION

To our knowledge, this is the first work to study neural simulators in terms of *test-time control of accuracy-cost trade-offs*. Since the performance varies with the chosen number of recurrent steps $K$, a scalar metric is no longer adequate; our experiments therefore focus on full accuracy-cost curve, and correlation-horizon surfaces. Across all tasks, the Recurrent-Depth Simulator provides a smooth, monotone trade-off, demonstrating that adaptive compute is possible, and we hope these results stimulate further work along this new axis.

Although the main experiments concentrate on RecurrSim instantiated with Fourier layers—chosen for their infinite receptive field (see Appendix C)—preliminary tests with convolutional blocks yield qualitatively similar results. We also use a recurrent-block with a single-layer for clarity: it delivers the most predictable behavior, however, deeper blocks also showed strong performance. Exploring richer blocks and alternative layer types under this controllable-compute paradigm remains a promising direction for future research.

