# OpenReview forum: "Test-Time Accuracy-Cost Control in Neural Simulators via Recurrent-Depth"
_ICLR.cc/2026/Conference — ICLR 2026 Poster_

### Official Review · Reviewer_YvLk · 2025-10-29

**Soundness:** 4
**Presentation:** 4
**Contribution:** 3
**Rating:** 8
**Confidence:** 4

**Summary:**

The paper introduce a new procedure for training any block of neural architectures when learning solutions of PDEs. This procedure consists of incorporating recurrent calls to the block, whose number is controlled by a parameter $K$. The parameter $K$ changes during the training to make the obtained reccurent network able to learn the solution for any $K$, with the intuition that the approximation will be more accurate for high $K$ than for low $K$. As a result, it is possible to tune the accuracy-cost trade-off at test time by toggling $K$. The approach is validated on several benchmark and for several underlying neural architectures.

**Strengths:**

- The paper is well written, easy and pleasant to follow
- The idea is simple, original, and represents a clever approach for adding an inductive bias towards physical solvers in the obtained neural network together with controlling the cost-accuracy trade-off.
- The approach is thoroughly validated on small to large scale physical learning problem, and its applicability to different existing SOTA architectures is demonstrated (RecurrFNO, RecurrVIT, RecurrUPT).

**Weaknesses:**

- The benchmark would benefit from a more systematic evaluation of UPT, ViT and FNO, i.e applying those tree models and their Recurr variants to all three high dimensional datasets.
- The high dimensional benchmark lacks a study on the effect of $K$.

**Questions:**

Are there practical limitations that prevented the authors to apply UPT, ViT and FNO and their Recurr variants to all three high dimensional datasets?

---

> ### Author Response · Authors · 2025-11-22
> **Response to Reviewer YvLk**
>
> We thank the reviewer for their feedback and positive evaluation of our work. We are grateful that the reviewer found our idea to be original and simple with a clever approach for adding an inductive bias, and recognized our thorough validation. We are especially grateful that the reviewer highlighted that our manuscript is well written, easy and pleasant to follow. We address each point below.
>
> > The benchmark would benefit from a more systematic evaluation of UPT, ViT and FNO, i.e applying those tree models and their Recurr variants to all three high dimensional datasets.
> > Are there practical limitations that prevented the authors to apply UPT, ViT and FNO and their Recurr variants to all three high dimensional datasets?
>
> We thank the reviewer for this excellent question. Yes, there are some practical limitations. Each example in the 3D Compressible Navier-Stokes dataset contains 262k grid points, resulting in extremely large activation tensors and memory requirements during training. Without modifications to the data (e.g., downsampling) or model (e.g., reducing depth), training ViT or UPT on this dataset would exceed our available GPU memory. We designed our ViT and UPT experiments to demonstrate 1) RecurrSim's plug-and-play property, and 2) RecurrSim's memory and parameter efficiency on different tasks.
>
> Full systematic comparisons across all combinations would require an estimated several weeks of GPU time. We are happy to add additional experiments to further strengthen the work and can provide preliminary results during the rebuttal period, with more polished results for the camera-ready version.
>
> > The high dimensional benchmark lacks a study on the effect of $K$.
>
> In Appendix I.3 (Figure 20), we study the effect of $K$ for two models trained on 3D Compressible Navier-Stokes with different training compute budgets ($\bar{K}$). We show that models trained with larger $\bar{K}$ achieve lower best trajectory error, but perform worse at low $K$.
>
> We have updated the manuscript to include a similar study for RecurrViT on Active Matter in Appendix I.3 (Figure 21). We will also add a similar study for RecurrUPT on ShapeNet-Car.

---

### Official Review · Reviewer_Pk3g · 2025-10-31

**Soundness:** 2
**Presentation:** 1
**Contribution:** 2
**Rating:** 2
**Confidence:** 4

**Summary:**

This paper proposes an architecture-agnostic framework, the Recurrent-Depth Simulator. During the training phase, the framework randomly samples the number of recurrent iterations K from a distribution and optimizes using truncated backpropagation; during the test phase, users can explicitly specify the number of iterations K to trade off between computational cost and simulation accuracy. The authors validate this framework across multiple datasets, including Burgers, Korteweg-de Vries (KdV), Kuramoto-Sivashinsky (KS), high-dimensional Compressible Navier-Stokes (CNS), Active Matter, and ShapeNet-Car. The method is compared against other adaptive-compute models, such as FNO-DEQ, ACDM, and PDE-Refiner, as well as standard architectures such as FNO, ViT, and UPT. The paper concludes that RecurrSim offers a superior accuracy-cost trade-off curve compared to baselines. On the high-dimensional CNS task, a lower-parameter RecurrFNO variant outperforms a higher-parameter FNO baseline while also reducing training memory.

**Strengths:**

- The framework's core contribution is providing explicit test-time control, allowing users to flexibly trade computational cost for accuracy by adjusting the number of iterations. Compared to baselines, this method offers a smoother, more predictable trade-off curve, avoiding the early saturation or erratic behavior seen in alternatives.
- The framework achieves excellent parameter efficiency through weight-sharing, enabling it to match or exceed larger baseline models with significantly fewer parameters and lower training memory consumption.
- The method is a plug-and-play, architecture-agnostic framework, and its generality has been validated across diverse backbones, including FNO, ViT, and UPT

**Weaknesses:**

- The core mechanism of this work,a recurrent-depth block trained with truncated backpropagation, is conceptually very similar to a standard Recurrent Neural Network (RNN), making the contribution potentially incremental as it applies existing techniques to a new domain.

- The paper is lacking in visual comparisons. The authors didn't provide corresponding visualizations for baselines like FNO-DEQ or ACDM, making it difficult to visually assess differences in physical fidelity. Also, none of the cases are provided with range.

- The paper suffers from several typographical errors and unclear phrasings. In particular, the descriptions of some experimental setups (e.g., Section 4.3 ) are brief, which may create difficulties for readers attempting to reproduce the results.

**Questions:**

See weaknesses

---

> ### Author Response · Authors · 2025-11-21
> **Response to Reviewer Pk3g**
>
> We thank the reviewer for their feedback. We are pleased that the reviewer recognized that our framework enables explicit test-time control, achieves excellent parameter efficiency, and is architecture-agnostic and plug-and-play. They also recognized that the framework has been validated across diverse backbones. We address each concern below.
>
> > The core mechanism of this work,a recurrent-depth block trained with truncated backpropagation, is conceptually very similar to a standard Recurrent Neural Network (RNN), making the contribution potentially incremental as it applies existing techniques to a new domain.
>
> **We did not claim that our contribution is a novel RNN architecture.** As the reviewer correctly commented in their Strengths section: our work's core contribution is providing explicit test-time control, allowing users to flexibly trade computational cost for accuracy by adjusting the number of iterations.
>
> We explicitly state our contributions here:
> 1) We introduce RecurrSim; an architecture-agnostic, plug-and-play framework that enables explicit test-time control over accuracy-cost trade-offs in neural simulators. This allows a single model to generate fast, less-accurate simulations for exploratory runs or real-time control loops; and more-accurate simulations for critical applications or offline studies.
> 2) We validate RecurrSim on diverse datasets (Burgers, Korteweg-De Vries, Kuramoto Sivashinsky, Active Matter, and ShapeNet-Car) with diverse backbones (FNO, ViT, UPT), demonstrating:
>     * Physically faithful simulations (even in low-compute settings);
>     * Memory and parameter efficiency on high-dimensional problems (RecurrFNO outperforms FNO with 13.5% less memory and 50% less parameters);
>     * True plug-and-play property via drop-in replacement in existing codes (ViT [1], UPT [2]).
> 3) We provide the first systematic study of test-time control in neural simulators. Unlike prior works, which focused only on final-iteration performance, we study the entire accuracy-cost curve. RecurrSim addresses key limitations of existing adaptive methods (FNO-DEQ, ACDM, PDE-Refiner): architecture dependence (FNO-DEQ by default, ACDM and PDE-Refiner require large backbones), early plateauing (all baselines plateaued at $K=4$), lack of *anytime prediction* capability (ACDM and PDE-Refiner require completion of the denoising process for a prediction) [3], poor generalization to OOD recurrent iterations (PDE-Refiner is dependent on parameter $K$), and parameter sensitivity (PDE-Refiner requires careful selection of noise schedule). This systematic study opens new research directions including adaptive stopping mechanisms and error estimation through recurrent iterations.
>
> **RecurrSim is not a traditional RNN (e.g., LSTM, GRU).** The reviewer may be conflating *recurrence through depth* with traditional *recurrent neural networks*. However, if by "standard RNN", the reviewer means the broad class of architectures that include weight-tied ResNets [4] and transformers [5], then yes, RecurrSim shares the property of weight-sharing across recurrent iterations. Still, this does not diminish our contribution of enabling test-time controllable neural simulators.
>
> **Our work is distinct from RNN use in prior works.** Li et al. [6] compare an FNO with temporal convolutions to an autoregressive FNO (which they call "FNO+RNN" but involves no hidden states or RNN parameters). Ren et al. [7] propose an autoencoder with a ConvLSTM for temporal propagation of the bottleneck. Wu et al. [8] propose a PINN with ConvGRU to model temporal relationships. Michałowska et al. [9] study the stabilization effects of combining neural operators with traditional RNNs (e.g., "simple RNN", LSTM, GRU). RecurrSim is distinctly different from all of these. It has no hidden states along the temporal dimension and does not use traditional RNNs.

---

> ### Author Response · Authors · 2025-11-21
> **Response to Reviewer Pk3g**
>
> > The paper is lacking in visual comparisons. The authors didn't provide corresponding visualizations for baselines like FNO-DEQ or ACDM, making it difficult to visually assess differences in physical fidelity. Also, none of the cases are provided with range.
>
> We thank the reviewer for this suggestion. We update the manuscript to include visualization comparisons for all baselines (FNO-DEQ, ACDM, and PDE-Refiner) in Appendix I.1 (Figure 16). We also update the manuscript to include each model's best performance across all recurrent iterations K (mean $\pm$ std) (Table 11):
>
> | Model | Burgers MSE ($\downarrow$) | KdV MSE ($\downarrow$) | KS Avg. Corr. Horizon $\alpha=0.8$ ($\uparrow$) | KS Avg. Corr. Horizon $\alpha=0.9$ ($\uparrow$) |
> | --- | --- | --- | --- | --- |
> | RecurrFNO  | $\mathbf{0.008 \pm 0.001}$ | $\mathbf{0.022 \pm 0.005}$ | $\mathbf{74.7 \pm 4.2}$ | $\mathbf{69.2 \pm 3.2}$ |
> | FNO-DEQ  | $0.045 \pm 0.007$ | $0.094 \pm 0.028$ | $41.9 \pm 5.6$ | $37.0 \pm 4.6$ |
> | ACDM  | $0.044 \pm 0.004$ | $0.043 \pm 0.015$ | $40.5 \pm 9.3$ | $35.6 \pm 9.2$ |
> | PDE-Refiner  | $0.161 \pm 0.024$ | $0.028 \pm 0.010$ | $61.6 \pm 7.7$ | $56.6 \pm 8.8$ |
>
> > The paper suffers from several typographical errors and unclear phrasings. In particular, the descriptions of some experimental setups (e.g., Section 4.3 ) are brief, which may create difficulties for readers attempting to reproduce the results.
>
> We note that Reviewers N727 and YvLk specifically comment "High-Quality Presentation: The paper is exceptionally clear, well-structured, and easy to follow. The appendices provide strong justifications for design choices." and "The paper is well written, easy and pleasant to follow." Nevertheless, we appreciate the feedback and will address any identified typographical errors.
>
> Regarding reproducibility, Appendices A-C provide a complete description of our hardware, data, data processing, hyperparameters, and optimization setups. We also provide complete source code for all experiments. We will further expand Section 4.3's experimental setups in the appendix to ensure complete clarity.
>
> [1]: Kohl, G., Chen, L.W. and Thuerey, N., 2023. Benchmarking autoregressive conditional diffusion models for turbulent flow simulation. arXiv preprint arXiv:2309.01745
>
> [2]: Alkin, B., Fürst, A., Schmid, S., Gruber, L., Holzleitner, M. and Brandstetter, J., 2024. Universal physics transformers. CoRR.
>
> [3]: Hu, H., Dey, D., Hebert, M. and Bagnell, J.A., 2018. Anytime neural network: a versatile trade-off between computation and accuracy.
>
> [4]: Liao, Q. and Poggio, T., 2016. Bridging the gaps between residual learning, recurrent neural networks and visual cortex. arXiv preprint arXiv:1604.03640.
>
> [5]: Feng, L., Tung, F., Hajimirsadeghi, H., Ahmed, M.O., Bengio, Y. and Mori, G., 2024. Attention as an RNN. arXiv preprint arXiv:2405.13956.
>
> [6]: Li, Z., Kovachki, N., Azizzadenesheli, K., Liu, B., Bhattacharya, K., Stuart, A. and Anandkumar, A., 2020. Fourier neural operator for parametric partial differential equations. arXiv preprint arXiv:2010.08895.
>
> [7]: Ren, P., Rao, C., Liu, Y., Wang, J.X. and Sun, H., 2022. PhyCRNet: Physics-informed convolutional-recurrent network for solving spatiotemporal PDEs. Computer Methods in Applied Mechanics and Engineering, 389, p.114399.
>
> [8]: Wu, B., Hennigh, O., Kautz, J., Choudhry, S. and Byeon, W., 2022, June. Physics informed RNN-DCT networks for time-dependent partial differential equations. In International conference on computational science (pp. 372-379). Cham: Springer International Publishing.
>
> [9]: Michałowska, K., Goswami, S., Karniadakis, G.E. and Riemer-Sørensen, S., 2024, June. Neural operator learning for long-time integration in dynamical systems with recurrent neural networks. In 2024 International Joint Conference on Neural Networks (IJCNN) (pp. 1-8). IEEE.

---

> > ### Comment · Reviewer_Pk3g · 2025-11-27
> >
> > I thank the authors for their response and the additional experiments. However, most of my concerns remain unsolved
> > 1. Novelty and model design. While the authors argue for a distinction between **recurrence through depth** and **temporal recurrence**, this distinction is semantic rather than algorithmic. The core mathematical formulation, $z_k = f(z_{k-1}, c)$, is identical to that of a standard RNN. Applying a recurrent structure to the problem of test-time compute control is indeed a valid application scenario. However, this represents a transfer of established techniques (RNNs + Weight Sharing + Time variance) to a new task, rather than a fundamental architectural innovation.
> > 2. Low quality of visualizations. The visualization in the revised paper fails to demonstrate the model's effectiveness.
> >  - Missing ground truth. The figure displays the predicted and absolute columns but omits the most critical ground-truth reference. Without the ground truth, it is impossible to assess the fidelity of the predictions.
> >  - Missing scales and colorbars. Figure 2, 13-16, lacks any colorbars or numerical indicators. It is unclear what the magnitude of the error is.
> >  - Lack of distinguishability. Visually, the predictions from the baseline models appear nearly identical to those of the proposed model. Without distinct visual differences or precise error scales, the qualitative comparison provides little value.
> >  Furthermore, the authors did not provide corresponding metrics for the specific samples visualized in the Navier-Stokes dataset, nor did they provide visual comparisons for other datasets. Consequently, the current visualizations do not support the claim that "RecurrSim generates physically faithful simulations" better than the baselines.
> >
> > After carefully reviewing the rebuttal and the revised paper, I will keep my score.
> >
> > Additionally, as a friendly suggestion, I strongly recommend that the authors polish all figures in the paper, including both the visualizations and the model architecture. Improving the aesthetic quality and clarity of these figures would significantly elevate the paper's presentation. That being said, I maintain a generally positive view of the paper's overall structure and organization.

---

> > > ### Author Response · Authors · 2025-11-28
> > > **Response to Official Comment by Reviewer Pk3g**
> > >
> > > > Low quality of visualizations. The visualization in the revised paper fails to demonstrate the model's effectiveness.
> > > > * Missing ground truth. The figure displays the predicted and absolute columns but omits the most critical ground-truth reference. Without the ground truth, it is impossible to assess the fidelity of the predictions.
> > >
> > > We respectfully disagree with this characterization. Figure 16 (old) shows predictions alongside absolute errors, computed as ground truth minus prediction. The absolute error directly quantifies deviation from the ground truth. Lighter colors indicate lower error and darker colors indicate higher error. **All error plots use the same scale for direct comparison.**
> > >
> > > We note that visually comparing predictions to ground truth would be uninformative here since most methods achieve trajectory errors in the $10^{-1}$ to $10^{-2}$ range, making predictions visually indistinguishable from ground truth. The absolute error plots are specifically designed to highlight these subtle differences. Nevertheless, we have updated the manuscript to include ground truth visualizations in Appendix I.1 (Figure 13) and Appendix I.2 (Figures 17-18) for completeness.
> > >
> > > > * Missing scales and colorbars. Figure 2, 13-16, lacks any colorbars or numerical indicators. It is unclear what the magnitude of the error is.
> > >
> > > We emphasize that all absolute error plots within each figure use identical colorscales, enabling direct visual comparison of error magnitudes across methods. However, to supplement Figure 2, we have updated the manuscript to include ground truth and colorbars in Appendix I.1 (Figure 13). Also, Figure 16 (old) now includes ground truth and colorbars in Appendix I.2 (Figures 17 and 18), and Figures 7 and 8 showing examples of the three-dimensional Navier-Stokes dataset now include colorbars.
> > >
> > > > * Lack of distinguishability. Visually, the predictions from the baseline models appear nearly identical to those of the proposed model. Without distinct visual differences or precise error scales, the qualitative comparison provides little value. Furthermore, the authors did not provide corresponding metrics for the specific samples visualized in the Navier-Stokes dataset, nor did they provide visual comparisons for other datasets. Consequently, the current visualizations do not support the claim that "RecurrSim generates physically faithful simulations" better than the baselines.
> > >
> > > The reviewer's observation that predictions "appear nearly identical" is expected. This is precisely why we focus on absolute error plots rather than side-by-side prediction comparisons. When methods achieve errors in the $10^{-1}$ to $10^{-2}$ range, predictions are visually indistinguishable from ground truth. We provided quantitative results demonstrating RecurrSim's performance compared to baselines.
> > >
> > > Regarding our claim that "RecurrSim generates physically faithful simulations (even in low-compute settings)": the reviewer appears to misinterpret this claim. Readers might expect that, in low-compute settings, RecurrSim would fail to reproduce key physical structures (shock in Burgers, solitons in KdV), or that errors would accumulate and produce unphysical artifacts or trajectory blow-up. However, in Section 4.2, we demonstrate RecurrSim correctly reproduces these key physical structures even in low-compute settings. This result directly supports our claim that "RecurrSim generates physically faithful simulations (even in low-compute settings)". Moreover, Figures 17 and 18, show that FNO-DEQ does not improve with additional compute and diffusion-based baselines perform worse precisely at the shocks and solitons. This directly validates our claim that "RecurrSim generates physically faithful simulations (even in low-compute settings)."
> > >
> > > > Additionally, as a friendly suggestion, I strongly recommend that the authors polish all figures in the paper, including both the visualizations and the model architecture. Improving the aesthetic quality and clarity of these figures would significantly elevate the paper's presentation. That being said, I maintain a generally positive view of the paper's overall structure and organization.
> > >
> > > We appreciate the reviewer's suggestion. However, without specific guidance on which aspects need improvement, it is difficult to act upon this feedback constructively. If the reviewer has specific concerns (e.g., Figure 1 doesn't show X), we would be happy to address them.
> > >
> > > We note that other reviewers rated presentation as "good" and "excellent", with Reviewers N727 and YvLk specifically commenting to praise the presentation.

---

> ### Author Response · Authors · 2025-11-28
> **Response to Official Comment by Reviewer Pk3g**
>
> > Novelty and model design. While the authors argue for a distinction between recurrence through depth and temporal recurrence, this distinction is semantic rather than algorithmic.
>
> We thank the reviewer for their continued engagement. However, we must respectfully point out that the reviewer's focus on whether an architecture falls into a broad category is entirely orthogonal to our contributions.
>
> Again, our contributions are:
> 1. **Framework that enables test-time control in neural simulators.** This is a standard capability in numerical methods but previously absent in neural simulators.
> 2. **Comprehensive empirical validation** demonstrating physically faithful simulations (even in low-compute settings), strong performance compared to existing adaptive baselines (FNO-DEQ, ACDM, PDE-Refiner), memory and parameter efficiency, and plug-and-play property.
> 3. **First systematic study of test-time control in neural simulators** demonstrating limitations of existing methods (e.g., early plateauing, lack of anytime prediction, poor generalization to OOD recurrent iterations) and establishing this as a new research direction.
>
> These contributions are independent of any taxonomic debate about what constitutes a "standard RNN."
>
> > The core mathematical formulation, $z_k = f(z_{k-1}, c)$, is identical to that of a standard RNN.
>
> For the sake of discussion, we note that the reviewer's claimed "standard RNN" formulation $z_k = f(z_{k-1}, c)$ is non-standard. Foundational RNN works [1, 2, 3, 4] define traditional RNNs as $z_k = f(z_{k-1}, c_k)$ with varying inputs $c_k$ (see Equation 1 in [1], Section 2.1 in [2], Equation 1 in [3], Equation 2.1 in [4]).
>
> In our previous response, we agreed that RecurrSim iteratively applies a function and, together with the weight-tied ResNet [5] and the transformer [2], can be broadly categorized as a *non-traditional* recurrent architecture. However, this class and reviewer's mathematical formulation, are so broad that they encompass all baselines in our paper (FNO-DEQ, ACDM, PDE-Refiner). These works were published at top venues and are recognized for their contributions despite also following iterative formulations. The reviewer's formulation would also classify any diffusion model and autoregressive LLM as "standard RNNs," rendering the term meaningless for evaluating novelty.
>
> In fact, all methods tested in our paper follow the mathematical formulation $z_k = f(z_{k-1}, c)$ along the depth dimension *and* $\mathbf{u}\_t = \mathcal{G}(\mathbf{u}\_{t-1}, p)$ along the temporal dimension. Therefore, every neural simulator satisfies the reviewer's mathematical formulation in at least the temporal dimension.
>
> > Applying a recurrent structure to the problem of test-time compute control is indeed a valid application scenario. However, this represents a transfer of established techniques (RNNs + Weight Sharing + Time variance) to a new task, rather than a fundamental architectural innovation.
>
> We designed RecurrSim to solve a practical problem, not to invent architectural primitives. By the reviewer's standard, many impactful works would be dismissed: Diffusion Models [6], FlashAttention [7], and LoRA [8] all apply existing techniques to enable new capabilities without claiming architectural innovation.
>
> [1]: Cho, K., Van Merriënboer, B., Gulcehre, C., Bahdanau, D., Bougares, F., Schwenk, H. and Bengio, Y., 2014. Learning phrase representations using RNN encoder-decoder for statistical machine translation. arXiv preprint arXiv:1406.1078.
>
> [2]: Feng, L., Tung, F., Hajimirsadeghi, H., Ahmed, M.O., Bengio, Y. and Mori, G., 2024. Attention as an RNN. arXiv preprint arXiv:2405.13956.
>
> [3]: Beck, M., Pöppel, K., Spanring, M., Auer, A., Prudnikova, O., Kopp, M., Klambauer, G., Brandstetter, J. and Hochreiter, S., 2024. xlstm: Extended long short-term memory. Advances in Neural Information Processing Systems, 37, pp.107547-107603.
>
> [4]: Yu, Y., Si, X., Hu, C. and Zhang, J., 2019. A review of recurrent neural networks: LSTM cells and network architectures. Neural computation, 31(7), pp.1235-1270.
>
> [5]: Liao, Q. and Poggio, T., 2016. Bridging the gaps between residual learning, recurrent neural networks and visual cortex. arXiv preprint arXiv:1604.03640.
>
> [6]: Ho, J., Jain, A. and Abbeel, P., 2020. Denoising diffusion probabilistic models. Advances in neural information processing systems, 33, pp.6840-6851.
>
> [7]: Dao, T., Fu, D., Ermon, S., Rudra, A. and Ré, C., 2022. Flashattention: Fast and memory-efficient exact attention with io-awareness. Advances in neural information processing systems, 35, pp.16344-16359.
>
> [8]: Hu, E.J., Shen, Y., Wallis, P., Allen-Zhu, Z., Li, Y., Wang, S., Wang, L. and Chen, W., 2022. Lora: Low-rank adaptation of large language models. ICLR, 1(2), p.3.

---

### Official Review · Reviewer_N727 · 2025-11-01

**Soundness:** 3
**Presentation:** 3
**Contribution:** 3
**Rating:** 8
**Confidence:** 3

**Summary:**

This paper introduces the Recurrent-Depth Simulator (RecurrSim), an architecture-agnostic framework designed to give neural simulators explicit, test-time control over their accuracy-cost trade-off. This capability is standard in classical numerical methods but largely absent in modern deep learning-based simulators.
The core idea is to replace a fixed-depth network with a recurrent block that is iterated a user-specified number of times (K) at inference. The model is trained by sampling K from a distribution and using truncated backpropagation-through-depth to maintain a fixed memory footprint. The authors demonstrate RecurrSim's effectiveness on a wide range of benchmarks and show that it can be applied to various backbones, consistently outperforming standard architectures and other adaptive-compute models.

**Strengths:**

1. Sufficient Novelty: The paper addresses a critical and practical problem in scientific machine learning. While the core mechanism (using recurrent iterations for an accuracy-cost trade-off) has been explored in other domains like computer vision and natural language processing, this paper's novelty lies in Its application and validation for the neural simulator domain, where this feature is a standard expectation from classical solvers but has been a major missing piece for deep learning methods.
2. Methodological Simplicity and Generality: The RecurrSim framework is "plug-and-play." It requires minimal code changes to a standard architecture – like LoRa methods. The paper strongly supports its "architecture-agnostic" claim by successfully applying it to FNO, ViT, and UPT.
3. Strong comparison with other baselines and in a wide range of PDE problems.
4. Scalability and Efficiency: The results on 3D CNS are impressive (a 0.8B param RecurrFNO outperforms a 1.6B param FNO with 13.5% less training memory).
5. High-Quality Presentation: The paper is exceptionally clear, well-structured, and easy to follow. The appendices provide strong justifications for design choices.

**Weaknesses:**

1. Lack of a Dedicated Reproducibility Section recommended in author guidelines (although appendix provides enough)
2. Insufficient Justification for truncated backpropagation-through-depth: The authors propose truncated backpropagation-through-depth to bound memory. However, they fail to discuss or compare this to gradient checkpointing, a standard alternative. Gradient checkpointing would compute the exact full-depth gradient (trading compute for memory) instead of the approximate gradient from truncated backpropagation-through-depth. The paper provides no justification for why an approximate gradient is sufficient or preferable.

**Questions:**

1. Minor comment Line 1215 Optimization typo. Can you correct?
2. You justify using truncated backpropagation-through-depth as a way to bound memory, which provides an approximate gradient. Could you elaborate on why this was chosen over gradient checkpointing, a standard alternative that computes the exact full-depth gradient by trading compute for memory?
3. The ICLR guidelines strongly encourage a dedicated 'Reproducibility Statement' paragraph to help reviewers locate the relevant details. While the appendices provide excellent, comprehensive details for reproducibility, this specific statement is missing. Would the authors be willing to add this paragraph in the final version to improve clarity for future readers?

---

> ### Author Response · Authors · 2025-11-21
> **Response to Reviewer N727**
>
> We thank the reviewer for their feedback and positive evaluation of our work. We are grateful that the reviewer found our work to contain sufficient novelty, methodological generality and simplicity,  and a strong experimental section. We are especially grateful that the reviewer highlighted our high-quality presentation. We address each point below.
>
> > Minor comment Line 1215 Optimization typo. Can you correct?
>
> We updated the manuscript to correct the typo on Line 1215 ("Optimzation" → "Optimization").
>
> > You justify using truncated backpropagation-through-depth as a way to bound memory, which provides an approximate gradient. Could you elaborate on why this was chosen over gradient checkpointing, a standard alternative that computes the exact full-depth gradient by trading compute for memory?
>
> We thank the reviewer for this excellent question. We chose truncated backpropagation-through-depth (TBPTD) over gradient checkpointing (GC) for two reasons: 1) TBPTD provides a sufficient approximate gradient while being computationally cheaper, and 2) TBPTD maintains more consistent batch processing times.
>
> 1) In Appendix D, we study backpropagation window $B$. Across all datasets, we find performance improves with $B$ but exhibits diminishing returns beyond $B=4$. This demonstrates that TBPTD with $B=4$ produces a sufficient approximate gradient for RecurrSim training. GC requires recomputing forward passes for discarded activations, adding $O(K)$ FLOPs overhead. This translates to real-world wall-clock time increase. TBPTD trades negligible amount of gradient accuracy for significant training speedup.
>
> |  | Forward FLOPs | Backward FLOPs | Recomputation FLOPs | Memory |
> | --- | -------- | --- | --- | --- |
> | Standard | $O(K)$ | $O(2K)$ | N/A | $O(K)$ |
> | Gradient Checkpointing | $O(K)$ | $O(2K)$ | $O(K)$ | $O(\sqrt{K})$ |
> | Truncated BP Through Depth | $O(K)$ | $O(2B)$ | N/A | $O(B)$ |
>
> 2) During training, $K$ is sampled from a distribution, so different examples in a minibatch may have vastly different $K$ values. TBPTD has constant backward cost $O(B)$ regardless of $K$, maintaining better model FLOPs utilization. With GC, examples with small $K$ finish earlier than those with large $K$, leading to poor model FLOPs utilization.
>
> > The ICLR guidelines strongly encourage a dedicated 'Reproducibility Statement' paragraph to help reviewers locate the relevant details. While the appendices provide excellent, comprehensive details for reproducibility, this specific statement is missing. Would the authors be willing to add this paragraph in the final version to improve clarity for future readers?
>
> We thank the reviewer for this suggestion. Following the ICLR 2026 Author Guide, we have updated the manuscript to include both, an Ethics Statement, and a Reproducibility Statement. We provide them below:
>
> **Ethics Statement.** We adhere to the ICLR Code of Ethics. Our work does not involve human subjects, create potentially harmful insights, raise discrimination/bias/fairness concerns, or pose privacy and security risks. Our work uses data generated from numerical solvers (which we make publicly available) and publicly available datasets. The aim of this work is to advance AI for Science. Although related methods may be deployed in critical applications, we do not deploy in such contexts. We have taken care to report results honestly, acknowledge limitations, and follow best practices for research integrity.
>
> **Reproducibility Statement.** We have made significant effort to ensure reproducibility. We use clear and consistent terminology throughout the work. Section 3 describes our framework in detail, explaining each design choice and linking to ablation studies in the Appendix. Section 4 describes the motivation and practical implementation of each experiment, with additional results in the Appendix. The Appendices provide complete descriptions of hardware specifications (Appendix A), data generation and processing (Appendix B), hyperparameters and optimization details (Appendix C), and pseudocode (Appendix J). We also provide source code for all experiments. Together, these resources enable independent researchers to reproduce and verify our findings.

---

### Official Review · Reviewer_XUFK · 2025-11-01

**Soundness:** 2
**Presentation:** 3
**Contribution:** 2
**Rating:** 4
**Confidence:** 3

**Summary:**

The authors propose a very simple framework for controlling, at test-time, the accuracy/speed of a neural simulator model, without requiring retraining or architecture adaptations. They show that this technique can be incorporated into a variety of architectures.

**Strengths:**

The proposed framework is easily used in multiple different architectures, and that flexibility is a strong point.

No additional custom losses or tricks are required, and the authors provide a simple explanation of the algorithm, making adoption simple.

The authors demonstrate improved performance over baselines with reduced compute/parameter counts.

**Weaknesses:**

The authors say that repeated applications of the recurrent block lead encourage the recurrent block to contract toward a fixed point — what is the justification for this claim? Is there any theoretical proof that the recurrent blocks do indeed converge toward a fixed point?

I would like to see some ablations on the initial latent distribution. The authors claim that the choice “primarily affects early iterations”. The authors also show that the early iterations are the ones that lead to the largest reduction in L2 error and are the most “important” in this sense, and so it would be interesting to see whether the choice of the initial latent distribution makes a big different in terms of overall performance of this method.

**Questions:**

Could the authors more clearly distinguish their method from DEQ, which also repeatedly applies a function (here the recurrent block) and converges to a fixed point, with the number of function applications being controllable to achieve a desired accuracy?

How was the recurrent iteration distribution chosen? It would be interesting to see how changing this distribution changes the performance of the model.

The authors show in the top of Figure 2 that performance saturates relatively quickly with the number of recurrent steps K, with the earliest steps leading to the largest reduction in L2 error. However, for memory purposes, the authors use a fixed backpropagation window where only the last B steps are backpropagated through, with the earlier steps being treated as constant. Would it not make more sense to backpropagate through the earliest recurrent layers, given that the earliest ones are the the ones that lead to the largest reduction in L2 error?

---

> ### Author Response · Authors · 2025-11-21
> **Response to Reviewer XUFK**
>
> We thank the reviewer for their feedback. We are pleased they found our framework flexible, simple to adopt, and performant. We address each point below.
>
> > The authors say that repeated applications of the recurrent block lead encourage the recurrent block to contract toward a fixed point — what is the justification for this claim? Is there any theoretical proof that the recurrent blocks do indeed converge toward a fixed point?
>
> We provide empirical evidence demonstrating that repeated applications of the recurrent block consistently lead the latent vector $z_k$ to converge toward a stable fixed point $z^* = z_K$.
>
> ---
>
> At test-time, the user is free to select the number of recurrent iterations $K$ according to their desired accuracy and available computational resources. Figure 2 (Top) and Figure 3 demonstrate that increasing recurrent iterations $K$ yields more accurate trajectories. To understand this behavior, we investigate how the latent vectors $z_k$ evolve with increasing recurrent iterations $K$ and whether they converge to fixed points $z^*$.
>
> **Experimental Setup.** We conduct experiments on two datasets: Burgers and long-horizon KdV. the trained RecurrFNO $\texttt{w/ EncDec}$ models from Section 4.2 with $K=32$. For each timestep in the trajectory, we measure the Euclidean distance between latent vector $z_k$ and approximate fixed point $z^* = z_K$ for all $k \in \left[0, K\right]$. We consider the convergence criterion $||z_k - z^*|| < \epsilon$ for a chosen tolerance $\epsilon$.
>
> **Results.** Figure 12 shows predicted trajectories for Burgers (Top Left) and long-horizon KdV (Top Right). The heatmaps (Bottom) show the Euclidean distance between latent vector $z_k$ and approximate fixed point $z^* = z_K$ for all $k \in \left[0, K\right]$ across time. For both equations, we see successive recurrent block applications generate increasingly similar latent vectors $||z_{k+1} - z^*|| < ||z_k - z^\*||$, eventually satisfying our convergence criterion $||z_k - z^\*|| < 1e^{-2}$. This indicates practical convergence to a fixed point. Remarkably, the convergence behavior reflects the underlying physics. For the Burgers equation, which exhibits two distinct regimes, an early nonlinear phase that develops shocks and a later viscous-dominated phase, we find that more recurrent iterations are required for convergence during the shock-formation period. This behavior matches that of adaptive numerical methods. For long-horizon KdV, we find spikes throughout the trajectory due to the more difficult task of soliton interactions. These results demonstrate that RecurrSim's latent vector does converge toward a fixed point and that the model behavior is aligned with the underlying physics.
>
> ---
>
> We update the manuscript to include this result in Appendix F.

---

> ### Author Response · Authors · 2025-11-21
> **Response to Reviewer XUFK**
>
> > I would like to see some ablations on the initial latent distribution. The authors claim that the choice “primarily affects early iterations”. The authors also show that the early iterations are the ones that lead to the largest reduction in L2 error and are the most “important” in this sense, and so it would be interesting to see whether the choice of the initial latent distribution makes a big different in terms of overall performance of this method.
>
> We perform additional experiments on the initial latent distribution. We investigate four different options: 1) the standard normal distribution, 2) a Dirac delta distribution at $\mathbf{0}$, 3) a Dirac delta distribution at the condition vector $\mathbf{c}$, and 4) a learnable initial latent vector. In agreement with Lines 245-250, we find the choice of initial latent distribution primarily affects early iterations, with minimal impact later as the latent vector converges toward the fixed point.
>
> ---
>
> RecurrSim's initial latent vector $\mathbf{z}_0$ is drawn from a fixed distribution $p(\mathbf{z})$ and iteratively updated through the recurrent-depth block. A good initial latent vector could potentially require fewer iterative updates to reach better latent vectors for decoding. In this experiment, we study how different initial latent distributions affect performance across varying recurrent iterations $K$.
>
> **Experimental Setup.** We conduct experiments on two datasets: Burgers and long-horizon KdV. We train a recurrent depth simulator with a point-wise lifting layer, a recurrent-depth block with a single Fourier layer, and a point-wise projection layer with $\sim$ 1M parameters. We set the backpropagation window $B=4$ and $\bar{K}=32$. We ablate four different initial latent vector choices:
> * Standard normal: $\mathbf{z}\_0 \sim \mathcal{N}(\mathbf{0}, \mathbf{I})$
> * Zero initialization: $\mathbf{z}\_0 = \mathbf{0}$ (Dirac delta at zero, $\delta\_\mathbf{0}$)
> * Conditioning initialization: $\mathbf{z}\_0 = \mathbf{c}$ (Dirac delta at conditioning vector, $\delta\_\mathbf{c}$)
> * Learnable initialization: $\mathbf{z}\_0 = \mathbf{z}\_\theta$ (learned parameter independent of input)
>
> After training, we evaluate the model at recurrent iterations $K \in \{2, 4, 8, 16, 32\}$ and report the trajectory error.
>
> **Results.** Table 8 and Table 9 show trajectory errors for all four initial latent vectors across different values of $K$. For Burgers equation, standard normal and zero initialization ($\delta_\mathbf{0}$) perform the best. At $K=2$, the learnable latent vector performs poorly, but by $K=16$, all initial latent vectors reach similar performance. For long-horizon KdV, zero initialization performs best, but we observe similar behavior: all initial latent vectors reach similar performance. Across both datasets and all recurrent iterations $K \in \{2,4,8,16,32\}$, we find zero initialization provides the most consistent performance. However, standard normal performs comparably and is consistent with DEQ and diffusion models. These results demonstrate while the initial latent distribution can play a part early on, given sufficient recurrent iterations, different distributions reach similar performance.
>
> Burgers Equation:
> | Recurrent Iteration $K$ | $\mathbf{z}\_0 \sim \mathcal{N}(\mathbf{0}, \mathbf{I})$ | $\mathbf{z}\_0 \sim \delta\_\mathbf{0}$ | $\mathbf{z}\_0 \sim \delta\_\mathbf{c}$ | $\mathbf{z}\_0 = \mathbf{z}\_\theta$ |
> | --- | --- | --- | --- | --- |
> | 2 | *0.3609* | **0.2183** | 0.5553 | 1.5921 |
> | 4 | *0.0968* | **0.0789** | 0.1287 | 0.1500 |
> | 8 | **0.0192** | *0.0199* | 0.0293 | 0.0205 |
> | 16 | *0.0101* | **0.0101** | 0.0132 | 0.0102 |
> | 32 | *0.0100* | **0.0099** | 0.0129 | 0.0102 |
>
> Long-horizon KdV Equation:
> | Recurrent Iteration $K$ | $\mathbf{z}\_0 \sim \mathcal{N}(\mathbf{0}, \mathbf{I})$ | $\mathbf{z}\_0 \sim \delta\_\mathbf{0}$ | $\mathbf{z}\_0 \sim \delta\_\mathbf{c}$ | $\mathbf{z}\_0 = \mathbf{z}\_\theta$ |
> | --- | --- | --- | --- | --- |
> | 2 | **0.5892** | 1.2367 | 1.8251 | *0.7051* |
> | 4 | 0.1570 | **0.1176** | 0.2412 | *0.1181* |
> | 8 | *0.0188* | **0.0167** | 0.0194 | 0.0201 |
> | 16 | 0.0187 | **0.0164** | *0.0176* | 0.0205 |
> | 32 | 0.0187 | **0.0164** | *0.0177* | 0.0205 |
>
>
> ---
>
> We update the manuscript to include this result in Appendix G.

---

> ### Author Response · Authors · 2025-11-21
> **Response to Reviewer XUFK**
>
> > Could the authors more clearly distinguish their method from DEQ, which also repeatedly applies a function (here the recurrent block) and converges to a fixed point, with the number of function applications being controllable to achieve a desired accuracy?
>
> The key difference between DEQs and RecurrSim, is that DEQs solve the direct problem, while RecurrSim performs truncated unrolling. [1]
>
> In the forward pass, DEQs train with a fixed point method (typically Anderson acceleration) with a set maximum iterations and tolerance on the residual. [2] Empirically, when we tested FNO-DEQ with Anderson acceleration, convergence failed for all reasonable tolerances, forcing maximum iterations every time. This explains the flat curves in Figures 3 and 15. FNO-DEQ cannot adapt to different recurrent iterations at test-time because it never was exposed to different recurrent iterations during training. On the other hand, RecurrSim samples the number of recurrent iterations $K \sim p(K)$, exposing the model to a wide range of recurrent iterations (computational budgets) for inference.
>
> In the backward pass, DEQs use implicit differentiation via the Implicit Function Theorem or estimates (e.g., phantom gradient).[3] Empirically, we found the phantom gradient to accelerate the backward pass and improve performance. RecurrSim simply uses truncated backpropagation-through-depth.
>
> > How was the recurrent iteration distribution chosen? It would be interesting to see how changing this distribution changes the performance of the model.
>
> The recurrent iteration distribution was chosen to concentrate samples around $\bar{K}$, while occasionally sampling much larger values. This strategy balances training FLOPs and performance across a wide range of recurrent iterations $K$.
>
> In Appendix E, we study how distribution parameter $\bar{K}$ affects performance across three datasets. In Figures 9-11, we show that larger $\bar{K}$ frequently achieves lower minimum trajectory errors, with diminishing returns beyond $\bar{K}=32$. However, models trained with larger $\bar{K}$ values underperform at small $K$ values (test-time). We chose $\bar{K}=32$ to balance performance across both low-compute and high-compute settings.
>
> > The authors show in the top of Figure 2 that performance saturates relatively quickly with the number of recurrent steps K, with the earliest steps leading to the largest reduction in L2 error. However, for memory purposes, the authors use a fixed backpropagation window where only the last B steps are backpropagated through, with the earlier steps being treated as constant. Would it not make more sense to backpropagate through the earliest recurrent layers, given that the earliest ones are the the ones that lead to the largest reduction in L2 error?
>
> Backpropagating through the earliest $B$ recurrent iterations is problematic because earlier iterations require gradients to flow backward from later iterations (See Lines 36-50 in Listing 2). Truncated backpropagation-through-depth ensures gradients flow from later iterations to the encoder.
>
> Note that, throughout training, all recurrent iterations $K \sim p(K)$ receive gradients. When small $K$ values are sampled, earlier iterations are backpropagated through, and when large $K$ values are sampled, later iterations are backpropagated through. In Appendix D, we study the effects of the backpropagation window $B$. Across all three datasets, increasing $B$ leads to better performance (for additional computational and memory costs), but with exponentially diminishing returns. On Burgers, increasing $B=4$ to $B=32$ provides minimal performance gain (0.0199 vs 0.0178) despite requiring 8x more memory.
>
> [1]: Bai, S., Kolter, J.Z. and Koltun, V., 2019. Deep equilibrium models. Advances in neural information processing systems, 32.
>
> [2]: Marwah, T., Pokle, A., Kolter, J.Z., Lipton, Z., Lu, J. and Risteski, A., 2023. Deep equilibrium based neural operators for steady-state pdes. Advances in Neural Information Processing Systems, 36, pp.15716-15737.
>
> [3]: Geng, Z., Zhang, X.Y., Bai, S., Wang, Y. and Lin, Z., 2021. On training implicit models. Advances in Neural Information Processing Systems, 34, pp.24247-24260.

---

### Author Response · Authors · 2025-12-03
**Summary of Reviews and Discussions**

We thank the area chairs and reviewers for their thoughtful feedback and time.

We are grateful that all four reviewers recognized our work's strengths: our work's novelty in addressing a critical problem in Scientific Machine Learning (N727, YvLk); our architecture-agnostic, plug-and-play framework for its ease of adoption and simplicity (XUFK, N727, Pk3g, YvLk); and our comprehensive experimental section demonstrating strong performance and parameter efficiency over existing methods (XUFK, N727, Pk3g, YvLk). We are especially pleased that Reviewers N727 and YvLk praised our presentation, noting the manuscript is "exceptionally clear, well-structured, and easy to follow" and "well written, easy and pleasant to follow."

We have addressed all reviewer concerns and strengthened the manuscript with the following additions: empirical convergence analysis demonstrating that repeated applications of the recurrent block lead the latent vector toward a fixed point (Appendix F, XUFK); ablation study on initial latent distributions (Appendix G, XUFK); Ethics and Reproducibility Statements (N727); explicit statement of our core contributions (Pk3g); additional plots with ground truth, absolute errors, and colorbars for all methods (Appendix I, Pk3g); and expanded study of K for high-dimensional datasets (Appendix I.3, YvLk).

---

### Meta-Review · Area_Chair_34id · 2026-01-08

**Summary:**

* Reviewers generally concern about comparison with other baselines both conceptually and empirically. For example, Reviewer YvLk is curious about comparison with UPT, ViT, and FNO. Reviewer Pk3g question about how this related to RNN. Reviewer N727 ask about the pros and cons comparing to using gradient checkpointing. Reviewer XUFK inquires about the positioning comparing to DEQ.
* Reviewers also request several additional ablation study to understand the design choice and hyperparameters.

**Reviewer Concerns:**

The authors provide empirical studies about fiex point analysis, ablation on initial latent, and a study of K in high-dimensional datasets. Authors also explains the difference and relations with prior works such as DEQ.  These results have addressed most of the major concerns.

**Reviewer Scores:**

I believe Reviewer N727 and YvLk will remain positive seeing the empirical results addressing their concerns as well as author's clarification in positions. Reviewer XUFK's concerns are also addressed mostly empirically, so I figure it's likely that XUFK might turn more positive after the rebuttal. I find it hard to agree with Reviewer Pk3g's position that the paper's contribution is not significant because the architecture can be written as an RNN. The authors' response is convincing by stressing both the relation and difference between the proposed framework and RNN, then noting that the key contribution is to enable test-time control of compute-accuracy trade-off in these scientific computing task.

---

### Decision · Program_Chairs · 2026-01-26

Accept (Poster)